# Stable hydrogen evolution reaction at high current densities via designing the Ni single atoms and Ru nanoparticles linked by carbon bridges

Rui Yao[1,6], Kaian Sun[2,6], Kaiyang Zhang[1], Yun Wu[1], Yujie Du[1], Qiang Zhao[1], Guang Liu ![ORCID][1] ✉, Chen Chen ![ORCID][3], Yuhan Sun ![ORCID][4,5] ✉ & Jinping Li ![ORCID][1,4] ✉

Continuous and effective hydrogen evolution under high current densities remains a challenge for water electrolysis owing to the rapid performance degradation under continuous large-current operation. In this study, theoretical calculations, operando Raman spectroscopy, and CO stripping experiments confirm that Ru nanocrystals have a high resistance against deactivation because of the synergistic adsorption of OH intermediates ($OH_{ad}$) on the Ru and single atoms. Based on this conceptual model, we design the Ni single atoms modifying ultra-small Ru nanoparticle with defect carbon bridging structure (UP-RuNi$_{SAs}$/C) via a unique unipolar pulse electrodeposition (UPED) strategy. As a result, the UP-RuNi$_{SAs}$/C is found capable of running steadily for 100 h at 3 A cm$^{-2}$, and shows a low overpotential of 9 mV at a current density of 10 mA cm$^{-2}$ under alkaline conditions. Moreover, the UP-RuNi$_{SAs}$/C allows an anion exchange membrane (AEM) electrolyzer to operate stably at 1.95 V$_{cell}$ for 250 h at 1 A cm$^{-2}$.

The overdependence on fossil fuels and resulting carbon emissions threaten the survival and development of global human society[1–3]. Hydrogen energy is becoming a significant source of clean energy by virtue of its effectiveness in conserving energy and reducing emissions. Hydrogen fuel cell vehicles are a prominent example of the application of this clean energy source, with hydrogen production playing a critical role in driving their adoption[4–6]. Electrochemical water splitting via hydrogen evolution reaction (HER) on the cathode is a promising and sustainable method for hydrogen production[7–10]. Despite the advancements in catalyst exploration thus far, there are remaining obstacles to achieving a viable hydrogen economy. Effectively, industrial water electrolysis must endure harsh conditions such as rapid charge/electron transfer at high current densities, robust

mechanical stability, bubble dynamics, and intermediate adsorption or coverage[11,12]. Essentially, accelerating the kinetics and improving the mechanical stability of HER at large current densities is a significant bottleneck in exploiting electrocatalysts for industrial hydrogen production[13,14].

Nowadays, although Pt-based catalysts are recognized to undergo the preeminent reaction kinetics in reducing the overpotential for HER, their limitations in availability and cost hinder large-scale implementation[15–18]. Ruthenium (Ru) has gained considerable attention as an affordable metal with the energy barrier for water splitting and hydrogen bonding strength comparable to Pt[9,19,20]. The combination of Ru clusters with single cobalt atoms anchored on nitrogen-doped carbon, as reported by Yuan et al., has demonstrated superior

[1]College of Chemical Engineering and Technology, Shanxi Key Laboratory of Gas Energy Efficient and Clean Utilization, Taiyuan University of Technology, Taiyuan 030024, China. [2]College of Materials Science and Engineering, Fuzhou University, Fuzhou 350108, China. [3]Department of Chemistry, Tsinghua University, Beijing 100084, China. [4]Shanxi Research Institute of Huairou Laboratory, Taiyuan 030031, China. [5]2060 Research Institute, Shanghai Tech University, Shanghai 201210, China. [6]These authors contributed equally: Rui Yao, Kaian Sun. ✉e-mail: liuguang@tyut.edu.cn; sunyh@sari.ac.cn; jpli211@hotmail.com

electrocatalytic HER activity. Such an outcome has been attributed to the alteration in the electronic structure and the Ru-H adsorption energy by the single cobalt atoms[21]. However, the inherent strong binding energy of Ru-H promotes the efficient adsorption of H, while for HER it is not always satisfactory[22,23]. In fact, the strong adsorption of OH intermediates on Ru frequently leads to the coverage of the active Ru sites, thus impeding the re-adsorption of water. Furthermore, there are relatively few reports on the regulation of Ru-OH adsorption energy for alkaline HER kinetics. Therefore, there is potential in adjusting the OH adsorption environment of Ru to facilitate continuous hydrogen evolution, especially at large current densities.

Herein, nickel (Ni) was chosen as the second base metal for designing catalysts because of its potential in catalyzing HER as a homologous transition metal element of Pt[24,25]. In our density functional theory (DFT) calculations, $Ni_{SAs}$, and Ni nanoparticles were separately introduced to compare their adsorption and thermodynamic effect of the OH intermediate on the Ru nanocrystalline matrix. Theoretical calculations indicate that $Ni_{SAs}$ have weaker adsorption for $OH_{ad}$ than Ni metal nanoparticles do. Therefore, incorporating $Ni_{SAs}$ into the Ru nanocrystalline system would significantly decrease the adsorption energy of Ru crystals for $OH_{ad}$. In addition, the carbon bridge effect of defective carbon promotes the redistribution of OH charges between Ni and Ru, weakening the adsorption of $OH_{ad}$ on Ru. Inspired by the aforementioned theoretical prediction, we employed a unipolar pulse electrodeposition (UPED) technique to place ultra-small Ru nanoparticles onto $Ni_{SAs}$ anchored defect carbon (UP-RuNi$_{SAs}$/C). Unlike continuous potentiostatic deposition in producing RuNi alloy nanoparticles (CP-RuNi/C), UPED effectively reduces the concentration polarization effect, allowing for precise control over the synthesis of small amounts of $Ni_{SAs}$ anchored on defect carbon through periodic pulse deposition. The optimized UP-RuNi$_{SAs}$/C electrocatalyst exhibits low overpotentials of 9 mV and 253 mV at 10 mA cm$^{-2}$ and 1000 mA cm$^{-2}$ respectively in 1.0 M KOH. Additionally, it retains its stability for 100 h at a higher current density

of 3000 mA cm$^{-2}$ without significant decline in activity. As expected, the AEM electrolyzer device assembled with the UP-RuNi$_{SAs}$/C catalyst as cathode exhibits a cell voltage of only 1.95 V at a current density of 1 A cm$^{-2}$. Our results here fully evidence the significance of accurately designing the catalyst structure to meet the demands of high-current water electrolysis.

## Results

### DFT calculation for catalyst screening and design

We commenced by conducting DFT calculations to investigate the impact of different Ni metal structures on the electronic features and the adsorption environment of Ru nanocrystals for HER. To model the structure of Ru nanocrystals, we focused on the Ru (002) surface of the hexagonal Ru structure (Supplementary Fig. 1a). Subsequently, Ni structures at various scales including Ni (111) index surface of cubic Ni (Supplementary Fig. 1b), Ni single atom anchoring on the substitutional (bulk) site of graphite carbon (Ni$_{sub}$/C) (Supplementary Fig. 1c) and Ni single atom anchoring on the edge (defect) site of graphite carbon (Ni$_{def}$/C) (Supplementary Fig. 1d) were constructed to investigate their effects on Ru nanocrystals. It is important to note that we performed an initial screening of the substrate carbon model in terms of formation energy, intermediate adsorption, and water decomposition barriers, respectively (Supplementary Figs. 3–8). We ultimately determined that the most stable and suitable Ni$_{SAs}$-anchored carbon defect is the zigzag-like graphene edge (Supplementary Fig. 3b). The adsorption of OH* was precisely investigated firstly through calculating the adsorption energy ($E_{OH^*}$) for different models in Fig. 1a and Supplementary Table 1. Notably, Ni$_{def}$/C has the weakest adsorption of OH* ($E_{OH^*}$ = 0.32 eV) compared to the Ni (111) of −0.25 eV and Ni$_{sub}$/C of 0.29 eV, and Ni$_{def}$/C has a superior free energy of H* ($|E_{H^*}|$) compared to Ni$_{sub}$/C, suggesting that Ni$_{SAs}$ is more favorable for weakening OH* adsorption on Ru sites relative to metal Ni, resulting in the optimal OH* adsorption energy for Ru (002) in RuNi$_{def}$/C ($E_{OH^*}$ = −0.15 eV) for exposure of active sites. In particular, in the RuNi$_{def}$/C model for this

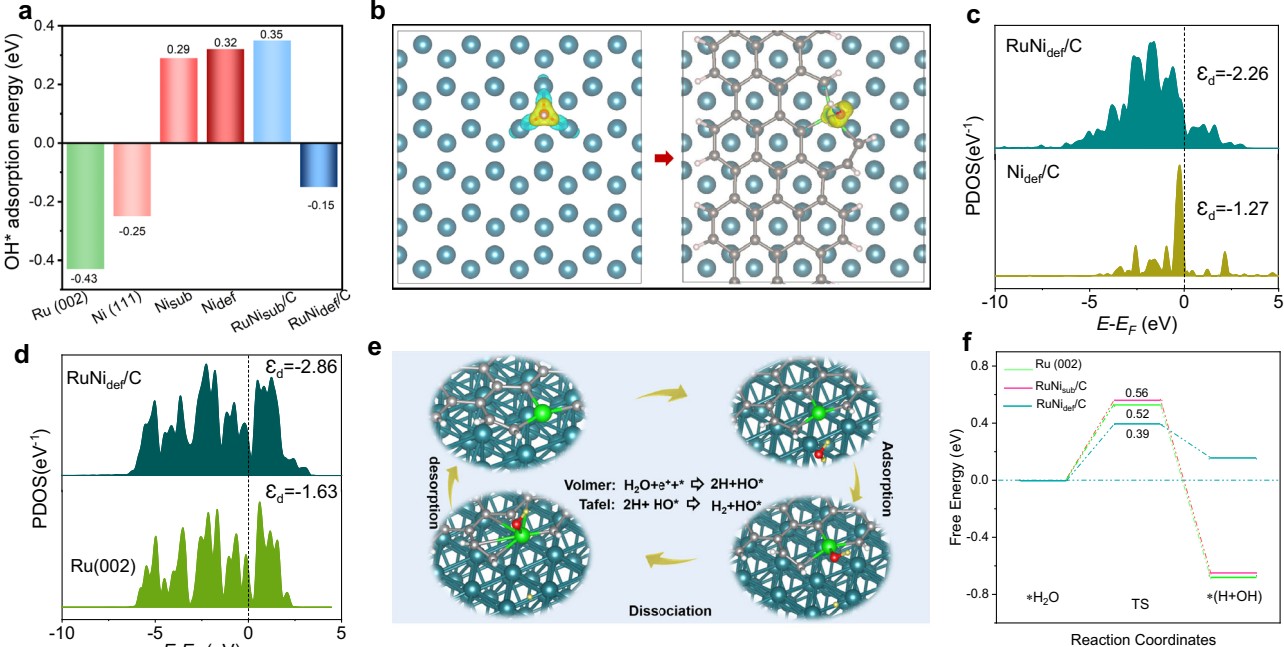

**Fig. 1 | Theoretical prediction results by DFT calculation. a** Adsorption energy of OH intermediate on active sites for HER corresponding to different models. **b** OH differential charge density calculation results for Ru (002) and RuNi$_{def}$/C models, in which color yellow indicates charge accumulation, and the color bright blue signifies charge depletion. **c, d** PDOS of Ni and Ru for Ni$_{def}$/C and RuNi$_{def}$/C. **e** HER reaction process for RuNi$_{def}$/C model under alkaline conditions. **f** Reaction coordinates for HER process for Ru (002), RuNi$_{sub}$/C, and RuNi$_{def}$/C models. Color codes for elements: carbon (gray), oxygen (red), hydrogen in water (yellow-green), hydrogen on the edge (white), nickel (green), and ruthenium (deep Blue).

work (Supplementary Fig. 3b), $Ni_{SAs}$ occupying the defective carbon sites, whereas there is no bonding between Ru and Ni. The elaborate electronic structure examine is firstly reflected in the OH differential charge density, for investigating the charge supply capacity of different structural models to OH group. Figure 1b and Supplementary Fig. 10 show that the OH-related charge redistribution of active Ru in the $RuNi_{def}/C$ model with introduction of Ni single atom anchored defect carbon, in which defective carbon serves as a carbon bridge (electron library) to establish OH charge connections between Ni and Ru sites, resulting in the synergistic adsorption of OH intermediates between the two sites. Therefore, the $RuNi_{def}/C$ model, with its reduced accumulation of charge on Ru by the modulation of $Ni_{def}/C$, ensures a weaker adsorption of OH than pure Ru (002) does. In addition, the partial density of states (PDOS) was used to examine the electronic interactions between the Ru and Ni sites and thus to analyze the causes of the OH adsorption energy changes. As shown in Fig. 1c, the introduction of $Ni_{def}$ into Ru caused a downward shift of the d-band center ($\varepsilon_d$) (−1.27 eV to −2.26 eV) and therefore caused the weakening of OH adsorption on the Ni site. While as the $Ni_{def}$ enters the pure Ru system, the d-band center shifts downward from −1.63 eV to −2.86 eV (Fig. 1d) due to the rearrangement of electrons around Ru, and the adsorption of Ru on OH is weakened. Notably, the electronic interactions between Ru and Ni weaken the adsorption of OH by $RuNi_{def}/C$, thus reducing the final water decomposition energy barrier.

Subsequently, we examined the predictable relationship between the above characteristics and the energy barrier of HER through DFT calculation. In the alkaline HER for $RuNi_{def}/C$, the initial adsorption of $H_2O$ molecules occurred selectively on the Ru site, as depicted in Fig. 1e and Supplementary Figs. 11–12. This Ru site is identified as the critical adsorption active site and active center through a poisoning experiment, as detailed in Supplementary Fig. 16. Following this initial adsorption, the $H_2O^*$ molecules underwent dissociation through the Volmer step, resulting in the production of $H_{ads}$ and $OH^*$. It is worth noting that the OH intermediate formed in the transition state is particularly adsorbed to the $Ni_{def}/C$ site (Supplementary Fig. 12b), demonstrating the synergistic effect of water decomposition between $Ni_{def}/C$ and Ru sites. Finally, the generated $H_{ads}$ underwent conversion into $H_2$ through the Tafel step and desorbed from the surface. According to the computational theory, Ru has strong adsorption capabilities for intermediates $OH^*$ or $H^*$ because of its effective water-splitting efficiency, which aligns well with our previous theoretical

prediction. Figure 1f illustrates that Ru (002) displays a transition state energy barrier of 0.52 eV close to that for $RuNi_{sub}/C$ (0.56 eV), indicating that $Ni_{sub}/C$ has a negligible effect on weakening the OH adsorption to lower the final water dissociation energy barrier. However, a low transition state energy barrier of 0.39 eV for $RuNi_{def}/C$ suggests that $Ni_{def}/C$ is optimal for regulating the adsorption energy of intermediates and accelerating water dissociation on active Ru, resulting in a notable advantage in the alkaline HER. It is noteworthy that the transition state energy barriers are positive for all models, which is consistent with the findings reported recently[20,26]. In addition, the RuNi alloy model has a higher transition state energy barrier of 0.7 eV (Supplementary Figs. 13–14), which does not reflect an advantage over pure Ru (0.52 eV) and $RuNi_{def}/C$ (0.39 eV), Further proving the unique modification effect of Ni single atom anchored defect carbon on Ru nanocrystals. However, it is worth noting that both RuNi alloy and $RuNi_{def}/C$ optimize H adsorption to some extent (Supplementary Fig. 15). To sum up, it is expected that the Ru and $Ni_{def}/C$ catalytic sites facilitate a synergistic adsorption of $OH^*$, in which the defective carbon as a link promotes synergistic water dissociation between the Ru and Ni sites and desorption of intermediates, leading to a reasonable alkaline HER activity.

## Material synthesis and characterization

Following these theories, we tried to assemble a Ni single atom modifying ultra-small Ru nanocrystalline structure. The main challenge in achieving this goal is managing rapid agglomeration and nucleation during solution synthesis, which ultimately hinder SAs formation. A unique UPED strategy was employed here to control the anchoring of $Ni_{SAs}$ on the carbon defect, ensuring the dispersion of $Ni_{SAs}$ for the modification of ultra-small Ru nanoparticles. Specifically, the reduction of $Ru^{3+}$ and $Ni^{2+}$ occurred when the current was turned on, with $Ru^{3+}$ being preferentially reduced. Disconnection of the current then allowed for diffusion of metal ion to the cathode, thereby maintaining its concentration recovery (Supplementary Figs. 17–18 and Fig. 2). The compensation gained by $Ru^{3+}$ during the off-stage weakens the overpotential of Ru electrodeposition, which effectively ensures the long-term deposition of Ru instead of Ni. At the final stage, Ni begins to be deposited and $Ni_{SAs}$ are formed. In contrast, the conventional potentiostatic approach generates an overpotential due to the concentration polarization by continuous consumption of $Ru^{3+}$, which causes $Ni^{2+}$ to reduce earlier and agglomerate into CP-RuNi/C alloy (Supplementary Fig. 19 and Fig. 2).

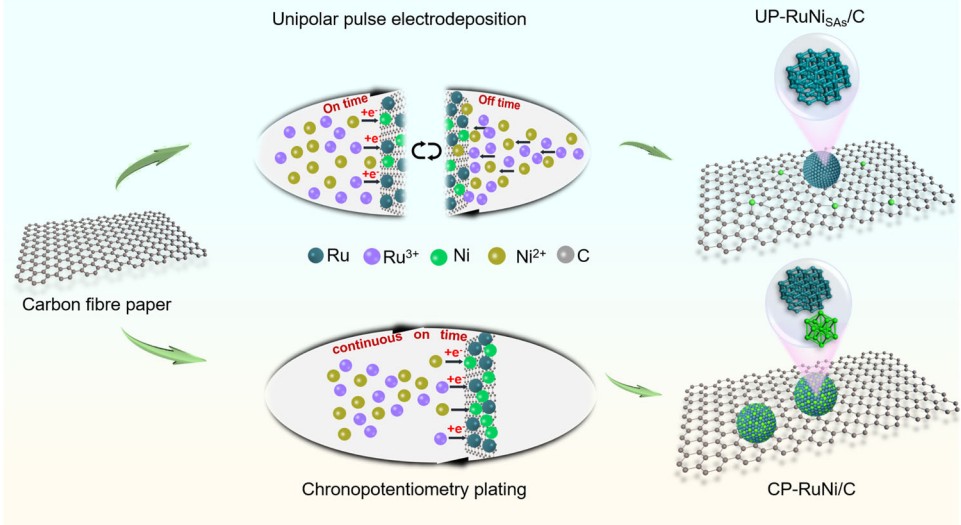

**Fig. 2 | Illustration of the synthesis procedure of UP-RuNi$_{SAs}$/C and CP-RuNi/C catalysts.** (I) UP-RuNi$_{SAs}$/C catalyst obtained by controlling intermittent current on-off states via a UPED strategy; (II) CP-RuNi/C catalyst obtain controlling current continuous conduction through a potentiostatic strategy.

The X-ray diffraction (XRD) patterns of the UP-RuNi$_{SAs}$/C and CP-RuNi/C (Fig. 3a and Supplementary Figs. 20–21) show the diffraction peaks corresponding to the Ru (PDF no. 06-0633), with the tiny particle size of Ru particles for UP-RuNi$_{SAs}$/C evident from the broad peaks. Particularly, the slow scanning XRD pattern of UP-RuNi$_{SAs}$/C shows diffraction peaks of Ru nanocrystals (PDF no. 06-0633) and there is no diffraction signal of Ni, which was initially thought to be a weak content of Ni or the presence of Ni$_{SAs}$. As indicated in Fig. 3b and Supplementary Fig. 22a of the scanning electron microscopy (SEM) images, agglomerate-like particles of about 2 μm in size were uniformly loaded on the carbon substrate, and further magnification (Supplementary Fig. 22b) showed that these particles of about 2 μm in size are agglomerated from smaller nanoparticles to form a rough surface. The transmission electron microscopy (TEM) image and the corresponding size distribution plot (Fig. 3c) further demonstrated that the size of the nanocrystalline is about 2 nm. As shown in Fig. 3d, the UP-RuNi$_{SAs}$/C displayed lattice fringes of 0.214 nm, corresponding to the (002) crystal plane of hexagonal Ru. Ni$_{SAs}$ and Ru nanocrystals can be accurately distinguished and identified using spherical aberration correction scanning TEM (AC-STEM). Figure 3e revealed the uniform dispersion of bright spots, indicating the uniform distribution of high-contrast single atoms on carbon fiber paper (CFP). Additionally, higher-resolution AC-TEM enables the observation of the scattered bright spots, while higher brightness Ru crystals can be seen due to the larger atomic number and the Ru (002) surface can be exposed (Fig. 3f). We can preliminarily speculate that the scattered bright spots around the Ru crystal are Ni$_{SAs}$, which was further validated through STEM-EDS. From Fig. 3g, it can be observed that Ru element is uniformly dispersed at the position of the nanocrystal, while Ni element is uniformly dispersed around the substrate and Ru. Inductively coupled plasma optical emission spectrometry (ICP-OES) accurately analyzed the Ru and Ni contents in the catalysts, revealing atomic ratios of 92.32/7.68 for Ru/Ni in UP-RuNi$_{SAs}$/C and 81.73/18.27 in CP-RuNi/C respectively (Supplementary Fig. 23). The CP-RuNi/C catalyst exhibited

a morphology (Supplementary Fig. 26a, b) resembling that of UP-RuNi$_{SAs}$/C. The average size of nanoparticles in CP-RuNi/C is about 5 nm, which was confirmed via Supplementary Fig. 26c. The HRTEM image of CP-RuNi/C revealed lattice fringes with 0.214 nm and 0.203 nm corresponding to the (002) and (111) planes of hexagonal Ru and cubic Ni, respectively, confirming the alloy configuration of the catalyst (Supplementary Fig. 26d). It is worth noting that the slow scanning XRD patterns of the CP-RuNi/C catalyst (Supplementary Fig. 20) only detected the diffraction peak of the Ru crystal. The corresponding diffraction peak of the Ni crystal was very weak or almost difficult to detect due to the lower content of Ni in the catalyst compared to Ru, which has been proven by ICP-OES (Supplementary Fig. 23). Moreover, pure Ru nanocrystals (UP-Ru/C) measuring approximately 3 nm in size (Supplementary Fig. 27) could also be prepared via the UPED strategy, indicating its consistency in controlling nanocrystal size.

We conducted X-ray absorption spectroscopy (XAS) at Ni K-edge of UP-RuNi$_{SAs}$/C to expound the single atom dispersion of Ni. Compared with the NiO, Ni$_2$O$_3$ and Ni-foil, the Ni K-edge X-ray absorption near-edge structure (XANES) spectra for UP-RuNi$_{SAs}$/C was markedly different with the edge of the Ni foil and NiO, with the white line between them, confirming its distinctive coordination structure (Fig. 4a). Generally speaking, d-p hybridization would occur for 3d transition metal elements like Ni when the coordination atoms lose their center symmetry, resulting in stronger white line intensities (H$_a$)[27,28]. The H$_a$ of Ni K-edge shows a raise to that of Ni foil, signifying the hybridization of Ni with heteroatom such as C of the CFP. As exhibited in Fig. 4b for the Fourier transformed $k^2$-weighted extended X-ray absorption fine structure (EXAFS) spectrum of UP-RuNi$_{SAs}$/C, no metal bonding peaks were found at 2–3 Å compared with NiO, Ni$_2$O$_3$, and Ni-foil, corroborating that the Ni is atomically dispersed and the absence of metal-metal coordination of Ni single atoms in UP-RuNi$_{SAs}$/C. It is worth noting that there is a bonding peak at 1.8 Å, and we speculated that it might correspond to a small amount of modified

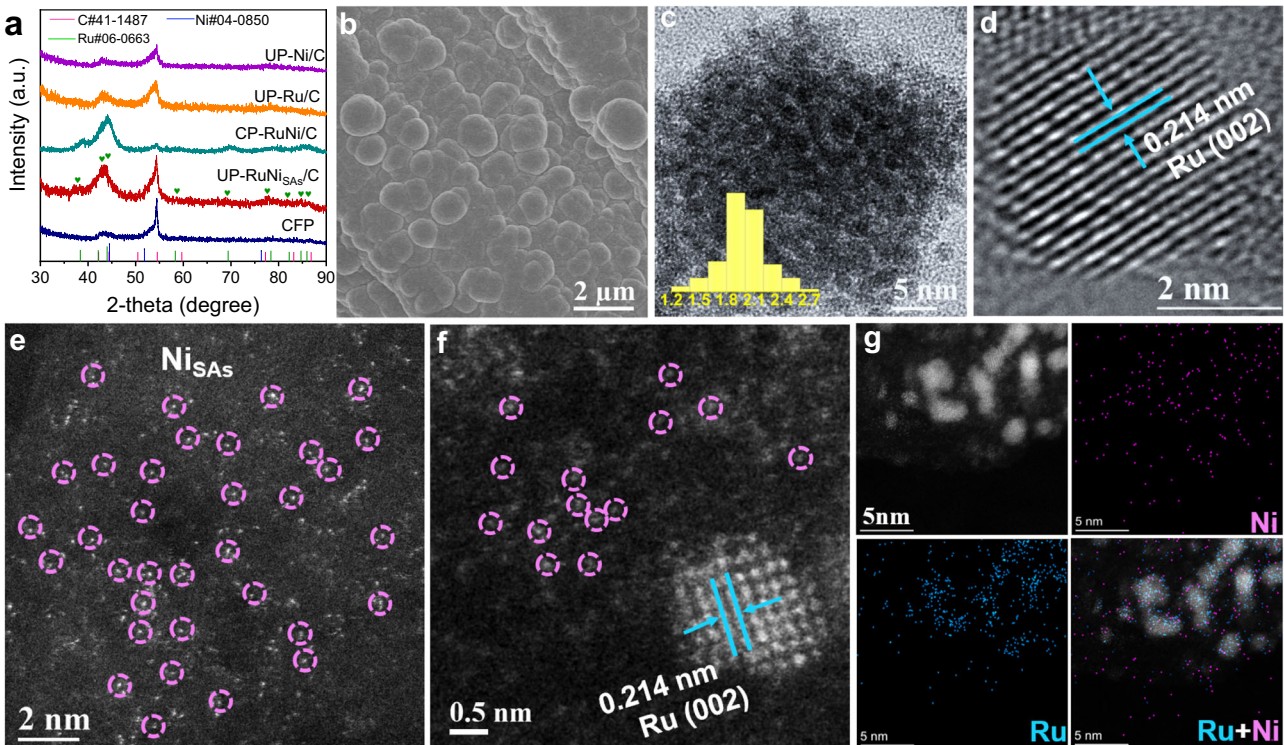

**Fig. 3 | Physical characterization and morphology of UP-RuNi$_{SAs}$/C. a** XRD patterns. **b** SEM image. **c** TEM image (inset: histogram of size distribution). **d** HR-TEM image. **e, f** Spherical aberration correction STEM, in which the bright spots highlighted by pink circles are ascribed to Ni single atoms. **g** STEM-EDS mapping of UP-RuNi$_{SAs}$/C.

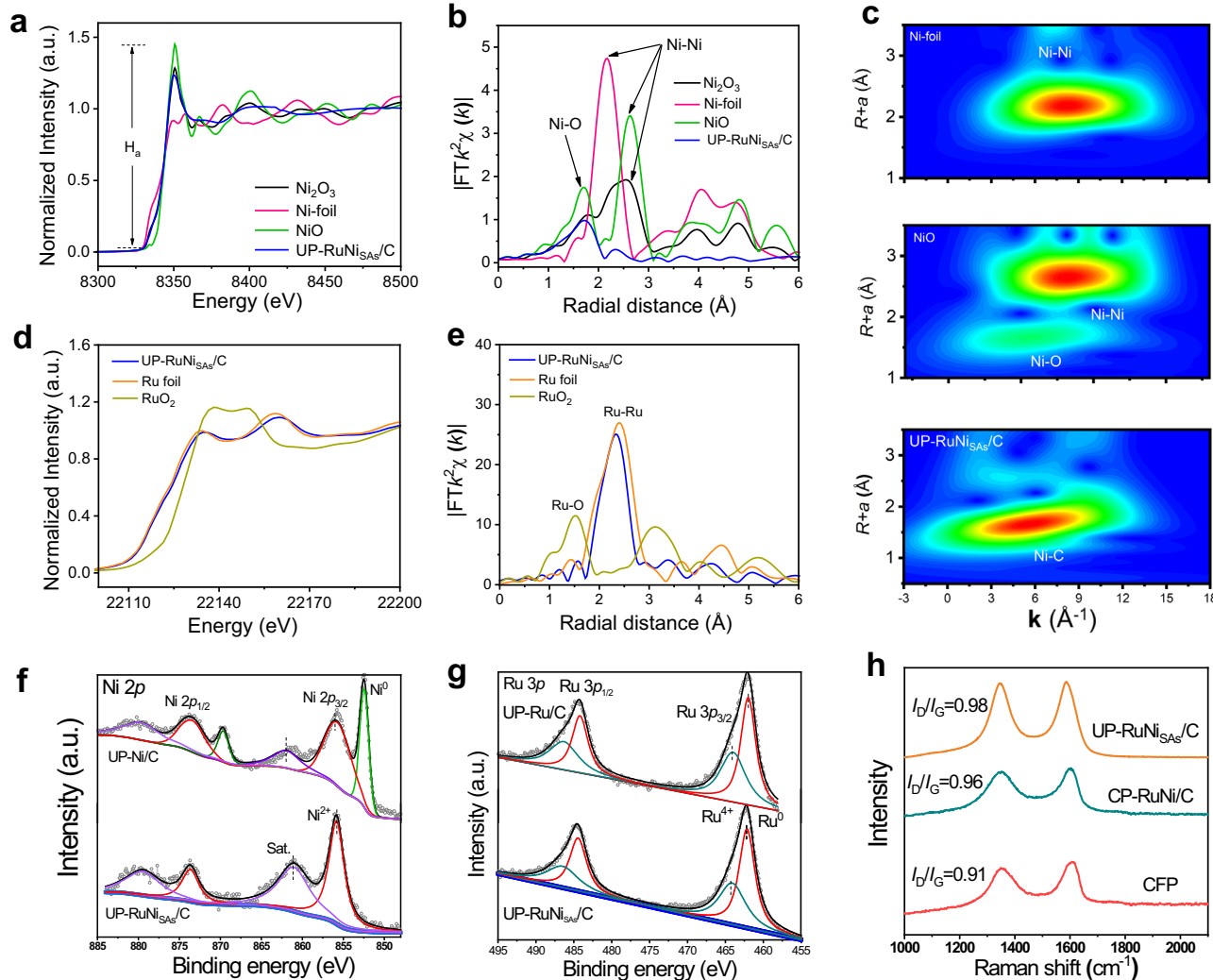

**Fig. 4 | Characterizations on chemical states and fine structures. a** Ni *K*-edge XANES analysis of UP-RuNi$_{SAs}$/C, NiO, Ni$_2$O$_3$, and Ni foil. **b** Fourier transforms EXAFS spectra at the Ni *K*-edge of UP-RuNi$_{SAs}$/C, NiO, Ni$_2$O$_3$, and Ni foil. **c** Wavelet transforms for the $k^2$-weighted EXAFS signs of UP-RuNi$_{SAs}$/C, NiO, and Ni foil. **d** Ni *K*-edge XANES analysis of UP-RuNi$_{SAs}$/C, Ru foil, and RuO$_2$. **e** Fourier transform EXAFS spectra of UP-RuNi$_{SAs}$/C, RuO$_2$, and Ru foil at Ru *K*-edge. **f** High-resolution XPS spectra analysis of Ni 2*p* for UP-Ni/C, UP-Ru/C, and UP-RuNi$_{SAs}$/C catalysts. **g** High-resolution XPS spectra analysis of Ru 3*p* for UP-Ni/C, UP-Ru/C, and UP-RuNi$_{SAs}$/C catalysts. **h** Raman analysis of CFP, CP-RuNi/C, and UP-RuNi$_{SAs}$/C catalysts.

oxidation or hybridization with carbon. The wavelet transforms (WT) of EXAFS at Ni *K*-edge were used to further investigate the coordination characteristics of UP-RuNi$_{SAs}$/C. For UP-RuNi$_{SAs}$/C, no Ni-Ni WT signal was observed in Fig. 4c unlike the Ni foil and NiO at ~8.2 Å$^{-1}$. In contrast, a Ni coordination with a non-metallic element was observed at ~6.0 Å$^{-1}$. Considering that the substrate of the catalyst is carbon, we speculate that it may be a Ni-C coordination. We verified the above supposition using the EXAFS fitting parameters at the Ni *K*-edge listed in Supplementary Fig. 29 and Supplementary Table 5. As expected, the coordination structure of Ni-C obtained by fitting the Ni$_{SAs}$ path is the optimal one and the coordination number (CN) is larger than that of Ni-O (which is caused by surface oxidation). It is worth noting that, as inferred from the absence of Ni-Ru bond in Fig. 4b and the EXAFS fitting in Supplementary Table 5, there is no bonding between Ni and Ru. The above results confirm that Ni is anchored to carbon in UP-RuNi$_{SAs}$/C.

To further explore the coordination of Ru, XAS at the Ru *K*-edge was recorded. Figure 4d, e displays the XANES and EXAFS for UP-RuNi$_{SAs}$/C, with Ru foil and RuO$_2$ as comparison samples. The Ru *K*-edge XANES spectra for UP-RuNi$_{SAs}$/C showed a similar position and trend to the Ru foil, but with a slightly positive shift in the near edge

and white line features (Fig. 4d). This suggests that Ru in the UP-RuNi$_{SAs}$/C was slightly oxidized in air. The existence of Ru-Ru coordination for RuNi$_{SAs}$/C is evident from the comparison with Ru foil, as UP-RuNi$_{SAs}$/C showed distinct Ru-Ru path at around 2.65 Å by the EXAFS results in Fig. 4e. The Ru-Ru WT signal of UP-RuNi$_{SAs}$/C (Supplementary Fig. 30) also demonstrated Ru-Ru coordination in Ru nanocrystals compared to Ru foil, with Ru-Ru coordination consistent with Ru foil specimens observed at ~9.8 Å$^{-1}$. The Ru-Ru coordination structure of the Ru element in RuNi$_{SAs}$/C is further evidenced by the EXAFS fitting parameters at the Ru *K*-edge in Supplementary Fig. 31 and Supplementary Table 6, and the fitting parameters revealed the decrease in the average CN of Ru in UP-RuNi$_{SAs}$/C compared to Ru foil, possibly due to the smaller size of Ru nanoparticles in the UP-RuNi$_{SAs}$/C catalysts (nano effect), and the increase in the number of surface atoms ultimately leads to insufficient number of coordinating atoms. It is preliminarily concluded that Ni single atoms modified Ru nanoparticles via carbon bridging (UP-RuNi$_{SAs}$/C) have been successfully synthesized, in which Ni single atoms are anchored to the substrate carbon.

The catalysts' surface properties were examined by X-ray photoelectron spectroscopy (XPS). As shown in Supplementary Fig. 32a, the survey XPS spectrum confirmed the presence of Ni, Ru, O, and C on the

surface of the UP-RuNi$_{SAs}$/C catalyst. The fitted Ni 2$p$ spectrum of the UP-RuNi$_{SAs}$/C (Fig. 4f) shows the valences of Ni$^{2+}$ at 856.1 eV and 873.9 eV due to surface oxidation[29]. Apparently, the Ni$^0$ peak was not exhibited in UP-RuNi$_{SAs}$/C, which is consistent with the AC-STEM results, where Ni is present in the form of single atoms. In contrast to UP-RuNi$_{SAs}$/C, the fitted Ni 2$p$ spectrum in UP-Ni/C displayed Ni$^0$ peaks at 852.51 eV and 869.71 eV. Nevertheless, the peak was shifted to lower binding energy compared with that of UP-Ni/C. The Ru$^0$ (462.15 eV and 484.13 eV) and Ru$^{4+}$(464.09 eV and 486.48 eV) peaks of UP-RuNi$_{SAs}$/C (Fig. 4g) positively shifted to the higher binding energy compared with that of UP-Ru/C, implying that electrons transferred from Ru to Ni for the UP-RuNi$_{SAs}$/C[30]. The C 1$s$ fitted peaks for the UP-RuNi$_{SAs}$/C (Supplementary Fig. 32b) were detected at 287.21 eV (C=O), 285.77 eV (C-OH), 284.87 eV (C-C) and 283.94 eV (C-Ni), which was consistent with the XAS results[24,31]. The Ru 3$d$ peaks were analyzed and fitted into Ru$^0$ and Ru-O. The shift of the Ru-O peak position to higher binding compared to UP-Ru/C was observed, indicating electron transfer from Ru to C in UP-RuNi$_{SAs}$/C. Accordingly, the electron transfer paths inside the UP-RuNi$_{SAs}$/C are Ru to Ni and Ru to C, constructing modified electronic structures of each element for improved activities. In addition, the fitted Ni 2$p$ spectrum of the CP-RuNi/C (Supplementary Fig. 33a) shows the valences of Ni$^0$ (at 852.7 eV and 869.9 eV), which is higher than that of reported for Ni$^0$ (852.5 eV) but similar to that of Ni in nickel carbides, implying the possible formation of C-Ni[24], which was supported by the C 1$s$ fitted peaks (Supplementary Fig. 33c) at 283.98 eV for the binding energy of C-Ni[24,31].

Moreover, the surface of the CFP was acidified-oxidative pretreatment (see the Supplementary Information for details) to introduce some oxygen-containing active groups (e.g., hydroxyl), which assists in anchoring the metal[32]. We recorded the Raman spectrum to detect the extent of carbon defect and the insertion of active sites for the CFP. As depicted in Supplementary Fig. 34, the $I_D/I_G$ value of electrochemical treatment CFP was significantly higher than that for the clean CFP, showing that the degree of carbon defects after pretreatment was significantly enhanced. Surprisingly, the scale of carbon defects for UP-RuNi$_{SAs}$/C was significantly fuller than that of CP-RuNi/C certified in Fig. 4h. Transient potentials under pulse conditions may more likely produce carbon defects, and the opulent defect is an ideal anchoring site for fixing metal ions to achieve atomic dispersion[33,34]. Note that during the UPED process, Ru is more likely to agglomerate into nanocrystals compared with Ni due to its higher cohesive energy, thereby allowing space for Ni$_{SAs}$ anchored on carbon[34–36].

## Electrochemical performance

The HER activities of UP-RuNi$_{SAs}$/C electrocatalyst at large current densities were investigated in 1.0 M KOH electrolyte. A saturated calomel electrode (SCE), used as the reference electrode, was calibrated in an H$_2$-saturated electrolyte (see Supplementary Fig. 35), and all the potentials here are reported with respect to the reversible hydrogen electrode (RHE). First, the linear sweep voltammetry (LSV) of the catalysts obtained by UPED under different conditions were summarized in Supplementary Fig. 36. The UP-Ru/C, UP-Ni/C, CP-RuNi/C, and Pt/C (20 wt%) materials were compared simultaneously. As shown in Fig. 5a, the UP-RuNi$_{SAs}$/C demonstrated the highest HER activity, with an overpotential of only 9 mV and 253 mV to deliver the current density of 10 mA cm$^{-2}$ and 1000 mA cm$^{-2}$ respectively, much smaller than that for commercial Pt/C (32 mV at 10 mA cm$^{-2}$). In addition, Ni$_{SAs}$-coupled ultrasmall Ru nanoparticles instead of RuNi bimetallic alloys yielded superior properties, as reflected by the overpotential of 33 mV at 10 mA cm$^{-2}$ for CP-RuNi/C with similar loading. The LSV curves without iR-compensated were provided in Supplementary Fig. 37, which is consistent with Fig. 5a that the optimal HER performance of UP-RuNi$_{SAs}$/C. To demonstrate the uniqueness of carbon substrate, we deposited the same UP-RuNi$_{SAs}$/Ti catalyst on titanium felt support using the UPED strategy. The results showed that the performance of

UP-RuNi$_{SAs}$/Ti[37,38] was still inferior to that of UP-RuNi$_{SAs}$/C (Supplementary Figs. 38–40), fully demonstrating the unique advantages of carbon sites in UP-RuNi$_{SAs}$/C. The Tafel slope of UP-RuNi$_{SAs}$/C was fitted as the minimum value of 37.6 mV/dec (Fig. 5b) among the UP-Ru/C, UP-Ni/C, CP-RuNi/C, and commercial Pt/C, indicating its most rapid HER kinetics. Importantly, the low Tafel slope of above Ru-based materials indicates that the rate-determining step of the reaction is the desorption of the OH intermediate, rather than the water decomposition[39]. Turnover frequency (TOF) is a critical descriptor for illustrating the intrinsic activity of catalysts. Therefore, the TOF values and number of active sites for the electrocatalysts were determined by the Cu-underpotential deposition (UPD) method (Supplementary Fig. 42a–e). As presented in Supplementary Fig. 42f, the TOF values of UP-RuNi$_{SAs}$/C at the overpotential of 20 mV and 50 mV were 1.40 H$_2$ s$^{-1}$ and 6.95 H$_2$ s$^{-1}$, much higher than that of other catalysts reported thus far in this work. The TOF values of UP-RuNi$_{SAs}$/C compares favorably with most catalysts in Fig. 5c and Supplementary Table 7, further proving the nice intrinsic activity of UP-RuNi$_{SAs}$/C. Based on the respective precious metal Ru mass loading at the overpotential of 20 mV and 50 mV, we further analyzed the normalized HER activity. The UP-RuNi$_{SAs}$/C exhibited the highest mass activity, with a value of 12.09 A per mg$_{Ru}$, which is 3.3 and 31.8 times higher than CP-RuNi/C and UP-Ru/C respectively, as shown in Fig. 5d.

Electrochemical impedance spectroscopy (EIS) obtained from the corresponding equivalent circuit diagram was utilized to explore the charge transfer capacity (Supplementary Fig. 43), in which the UP-RuNiSAs/C exhibited the lowest charge transfer internal resistances ($R_{ct}$) of 0.10 Ω and 0.12 Ω. (Supplementary Table 8), indicating the optimum conductivity of UP-RuNi$_{SAs}$/C towards the HER. Electrochemically active surface area (ECSA) can be used as an indicator to assess the number of active sites of catalysts. As expected, the calculated double layer capacitance ($C_{dl}$) value (Supplementary Fig. 44) of UP-RuNi$_{SAs}$/C according to the cyclic voltammetry (CV) curves at different scan rates (Supplementary Fig. 45) proved to be the highest (292.06 mF cm$^{-2}$), leading to the higher ECSA value of 584.5 m$^2$ g$^{-1}$ compared with the CP-RuNi/C (353.88 m$^2$ g$^{-1}$) and Pt/C (399.22 m$^2$ g$^{-1}$), demonstrate the ability of UP-RuNi$_{SAs}$/C to provide more active sites for boosting the HER performance. From the perspective of long-term stability and multiple cycle CV test, the extraordinary superiority of UP-RuNi$_{SAs}$/C for practical application at large current densities was further corroborated. As revealed in Supplementary Fig. 46, the LSVs of UP-RuNi$_{SAs}$/C before and after thousands of cycles CV demonstrate no significant changes, even when subjected to the relatively high current density. This confirms the practical applicability of UP-RuNi$_{SAs}$/C, especially at large current densities. The long-term stability of industrial water electrolysis is largely limited by the current density exceeding 500 mA cm$^{-2}$. The UP-RuNi$_{SAs}$/C was subjected to chronopotentiometry test at a current density of −10 mA cm$^{-2}$ and retained absolute stability for 200 h, superior to the commercial Pt/C electrocatalyst with the continuous decline in stability (Supplementary Fig. 47). Notably, the UP-RuNi$_{SAs}$/C also gave a stability for 100 h under high current densities ranging from −200 mA cm$^{-2}$ to −3000 mA cm$^{-2}$ (Fig. 5e and Supplementary Fig. 48), demonstrating the potential for efficient and continuous response in industrial hydrogen production. In addition, the amount of H$_2$ produced during the first 3 h was determined via gas chromatography (Supplementary Fig. 49), demonstrating that its faradaic efficiency was nearly 100%. The SEM (Supplementary Fig. 50) and XPS (Supplementary Fig. 51) after the electrochemical test further proved its considerable stability, with the morphology and chemical state of major elements barely changed. Moreover, the UP-RuNi$_{SAs}$/C also presented favorable HER performances in the acidic and neutral conditions. As shown in Supplementary Fig. 52, the UP-RuNi$_{SAs}$/C behaved the best activity in 0.5 M H$_2$SO$_4$ and 1.0 M phosphate buffer solution (PBS), with the smallest overpotentials of 18 mV and 27 mV, respectively, and considerable

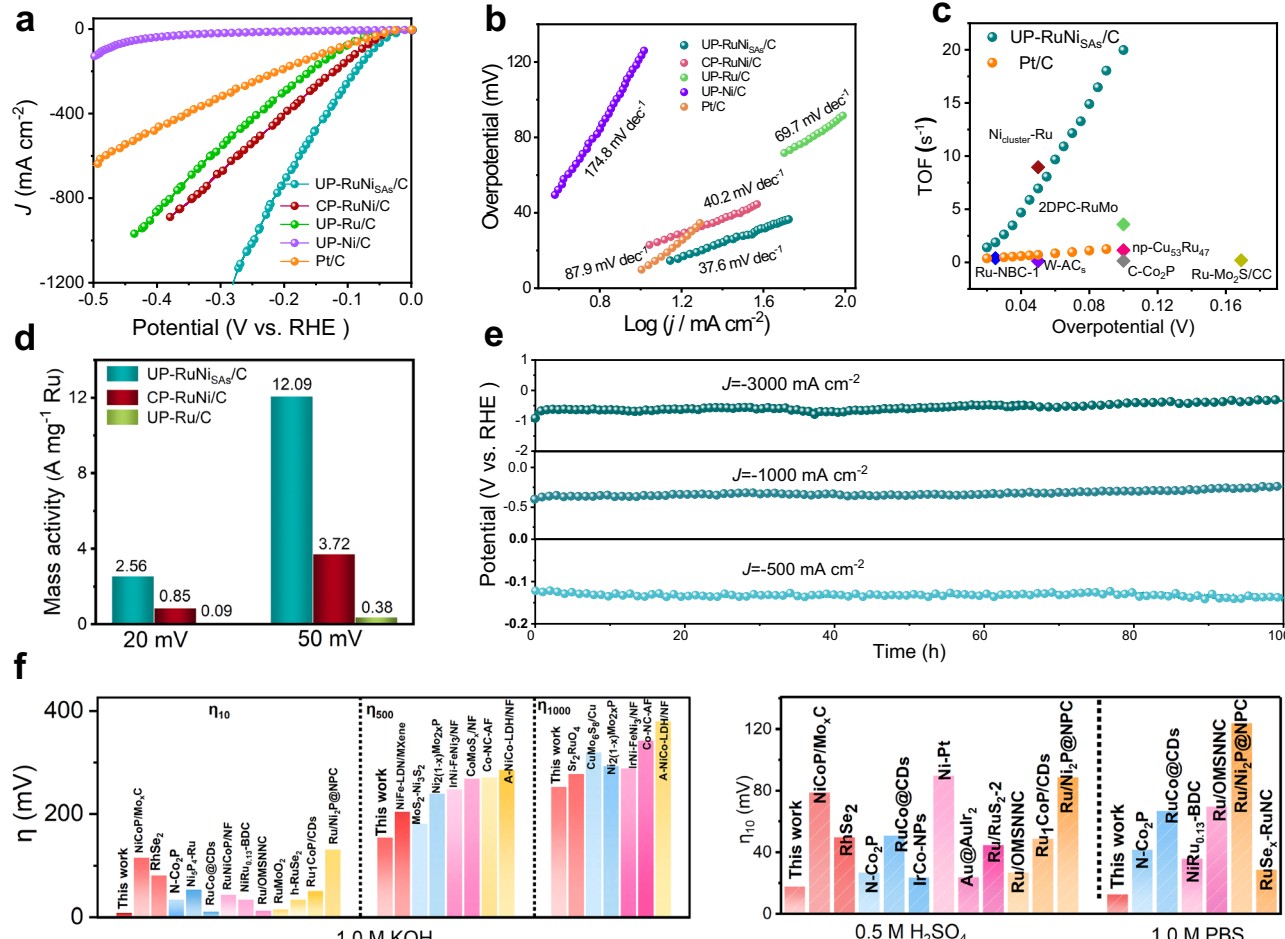

**Fig. 5 | Electrochemical performances. a, b** Polarization curves and Tafel plots of the UP-Ni/C, UP-Ru/C, CP-RuNi/C, UP-RuNi$_{SAs}$/C and commercial Pt/C catalysts in 1.0 M KOH. **c** TOF values of UP-RuNi$_{SAs}$/C compared with Pt/C and previously reported catalysts in 1 M KOH. **d** Activity of Ru mass normalization of the UP-Ru/C, CP-RuNi/C, UP-RuNi$_{SAs}$/C catalysts at $\eta_{20}$ and $\eta_{50}$ in 1 M KOH. **e** Long term stability testing of the UP-RuNi$_{SAs}$/C catalysts in 1.0 M KOH. **f** Comparison in terms of overpotentials for UP-RuNi$_{SAs}$/C and recently reported electrocatalysts at different current densities ($\eta_{10}$, $\eta_{500}$ and $\eta_{1000}$) in 1.0 M KOH and pH-universal performance comparison of UP-RuNi$_{SAs}$/C and recently reported HER electrocatalysts. The catalyst loading was 1.25 mg cm$^{-2}$, the pH of the electrolyte was 13.6, and the liquid resistance was about 1.613 Ω.

cycling and long-term stability of 28 h compared with the Pt/C (Supplementary Fig. 53b and Supplementary Fig. 55b). Finally, its pH-universal and high-current-density HER activity superior to those of other reported electrocatalysts are summarized in Fig. 5f and Supplementary Table 11. All the above results demonstrate the remarkable activity of UP-RuNi$_{SAs}$/C at large current densities for HER and the tremendous application foreground.

To investigate the application prospect of UP-RuNi$_{SAs}$/C as a cathode material under actual industrial conditions, we constructed an electrolytic cell based on an anion exchange membrane. We used the commercial NiFeO$_x$ catalyst as the anode and compared the performance of UP-RuNi$_{SAs}$/C with the Ru/C and Pt/C catalysts separately at the cathode commercially[40]. The AEM electrolytic cell was constructed into a sandwich configuration with an area of 1 cm$^2$, consisting of a current collector, cathode (anode) catalyst, and anion membrane without hot pressing, as shown in Fig. 6a. As we expected, the performance of AEM electrolyzer based on UP-RuNi$_{SAs}$/C catalyst as cathode was improved as the temperature rose (Fig. 6b). Surprisingly, low cell voltage of 1.70 V and 1.95 V were recorded at 0.5 A cm$^{-2}$ and 1 A cm$^{-2}$ respectively at 70 °C for UP-RuNi$_{SAs}$/C catalyst, whereas the Ru/C and Pt/C and catalysts obtained the cell voltage of 1.91 V and 2.27 V separately at 500 mA cm$^{-2}$ (Fig. 6c and Supplementary Fig. 57). More importantly, the energy conversion efficiencies for the

electrolytic cell based on UP-RuNi$_{SAs}$/C and Pt/C catalysts are 76.46% and 44.67% (Fig. 6d) at 0.5 A cm$^{-2}$. To further verify its practical application potentials in nearly the industrial scale, the long-term stability of the electrolyzer was studied for nearly 250 h at 1 A cm$^{-2}$. As revealed in Fig. 6e and Supplementary Fig. 58, the UP-RuNi$_{SAs}$/C gave a great long-term stability with negligible potential fluctuations compared with the Pt/C and Ru/C catalysts, suggesting the promising prospect of UP-RuNi$_{SAs}$/C for hydrogen production.

## Kinetics and reaction mechanism of the UP-RuNi$_{SAs}$/C for alkaline HER

As mentioned above, the low Tafel slope confirms that rate-determining step of the reaction is the desorption of the OH intermediate. To gain further in-depth understanding on the reaction mechanism of UP-RuNi$_{SAs}$/C toward alkaline HER and to observe the specific adsorption (desorption) behavior of OH intermediate, we conducted CO-stripping experiment and operando Raman spectroscopy. Electrochemical oxidation of CO is triggered by adsorption of active OH, and thus the negative shift of the oxidation potential is generally considered indicative of a stronger OH affinity for the catalyst[41]. As shown in Fig. 7a, the UP-RuNi$_{SAs}$/C demonstrated the lowest OH adsorption capacity with the peak potentials of 0.74 V compared with the pure UP-Ru/C (0.61 V), implying the better OH

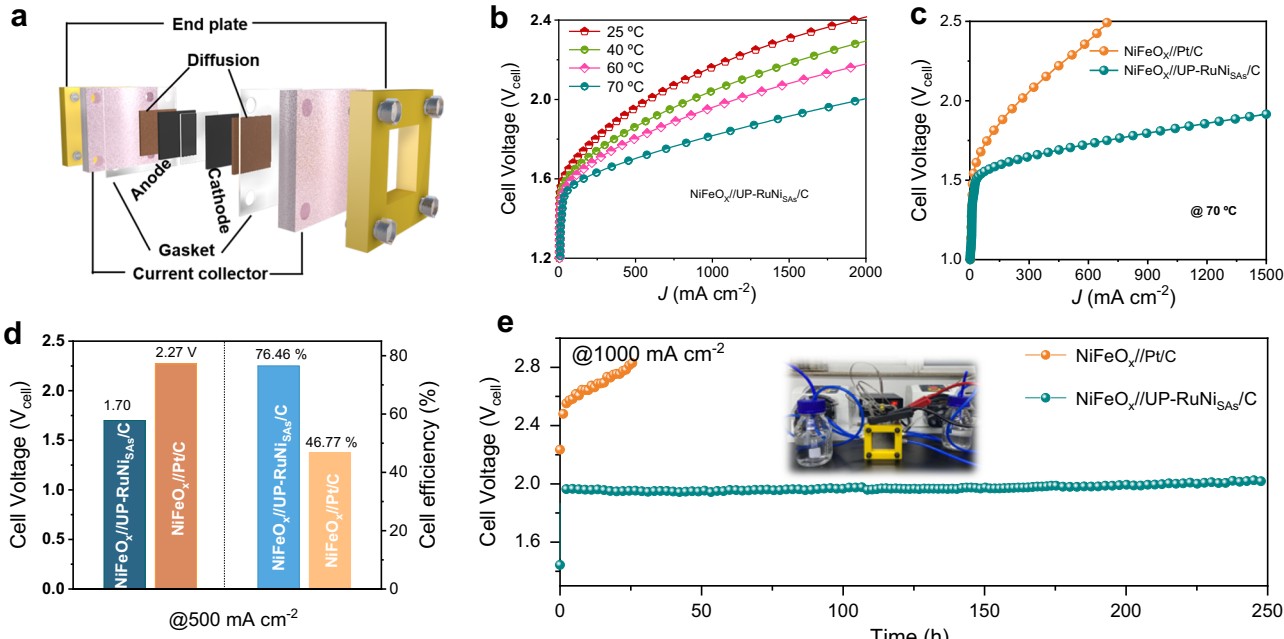

**Fig. 6 | Alkaline HER performance in the AEM reactor. a** Schematic illustration of the AEM reactor. **b** LSVs operated at different cell temperatures in an AEM electrolyzer. **c** LSV curves of the AEM reactors using NiFeO$_x$ and UP-RuNi$_{SAs}$/C, Pt/C catalysts as the anodic and cathodic electrodes, respectively at 70 °C. **d** Cell voltage and energy efficiency of the AEM electrolyzer at 0.5 A cm$^{-2}$. **e** Stability tests of the AEM water electrolyzers at 1 A cm$^{-2}$. The loading capacity of the catalyst is 1.25 mg cm$^{-2}$, the pH of the electrolyte was 13.6, and the liquid resistance was about 0.0902 Ω.

desorption kinetics of Ru site in UP-RuNi$_{SAs}$/C with the introduction of Ni$_{SAs}$ during the Volmer step. The more negative peak potential of CP-Ru/C and CP-RuNi/C than that of RuNi$_{SAs}$/C suggests that the higher OH adsorption capacity of CP-RuNi/C in Fig. 7b, which causes a decrease in water re-adsorption for the Ru site in CP-RuNi/C.

Operando Raman spectroscopy was conducted to explore the changes of surface species during alkaline HER. As presented in Fig. 7c and Supplementary Fig. 59, the OH species (at 1042 cm$^{-1}$) emerged and vanished earlier at 0.6 V vs. RHE and −0.1 V vs. RHE for UP-RuNi$_{SAs}$/C catalysts compared with UP-Ni/C (0.1 V vs. RHE and −0.6 V vs. RHE) and UP-Ru/C (0.4 V vs. RHE and −0.3 V vs. RHE) when the gradually reduced potentials were applied, implying the Ni$_{SAs}$ assisted Ru nanoparticles to produce OH species via water decomposition and OH desorption at a relatively positive potential[42]. For CP-RuNi/C in Fig. 7c, the relatively negative potential for OH occurrence (0.4 V vs. RHE) and the OH disappearance (−0.2 V vs. RHE) further proved its inferior kinetic process for water splitting and OH desorption to that of Ni$_{SAs}$-decorated Ru nanoparticles, and confirmed the distinctive contribution of Ni$_{SAs}$ for enhance the desorption effect of OH intermediates on Ru nanocrystal surfaces. Furthermore, the rapid desorption of bubbles at high current densities directly affects the mechanical stability of the electrode, the interaction between the catalyst and electrolyte, and ultimately affects the overall mass transmission. Rapid bubble release plays a key role in these processes[39]. Therefore, we investigated the hydrophilicity of different materials through contact angle test. As shown in Supplementary Fig. 60, the UP-RuNi$_{SAs}$/C electrocatalyst gave the smallest contact angle of 49.4°, significantly lower than that of UP-Ru/C (99.5°) and CP-RuNi/C (67.0°), proving the suitable hydrophilicity of UP-RuNi$_{SAs}$/C for favorable formation of small bubbles and the release of bubbles for mass transfer.

Overall, Fig. 7d illustrates the detailed process of the entire catalytic system for promoting water decomposition and to hydrogen production. On the one hand, Ni$_{SAs}$ promote the rapid water dissociation of Ru to produce OH intermediates; on the other hand, due to the weak adsorption of Ni$_{SAs}$ on OH intermediates, Ni$_{SAs}$ are synergistically coupled to Ru with optimized kinetics in the desorption of OH intermediates, which prevents the strong adsorption of OH intermediates by Ru. while this process is achieved through the carbon matrix as a carbon bridge, thus allowing the thermodynamic and kinetic dramatic cyclic response between the two sites.

## Discussion

We reported the Ni single atom modifying ultra-small Ru nanoparticle structure (UP-RuNi$_{SAs}$/C) for continuous high-current-density and pH-universal hydrogen production. The catalyst could run stably for 250 h in a practical AEM electrolyzer. Anchored on carbon defects, Ni$_{SAs}$ play a crucial role in HER process, leading to an optimized adsorption kinetics of intermediates at the active site of Ru nanoparticles. This results in the effective regulation of adsorption energy and electrons environment. Specifically, the continuous re-adsorption of water molecules at the active site is ensured through synergistic adsorption of OH intermediates with Ni$_{SAs}$. Consequently, this guarantees the continuous and efficient generation of hydrogen at high current densities. Our efforts here will not only provide a novel approach to the coordinated design of single-atom sites for effective HER catalytic systems in practical AEM electrolyzers but also highlight the importance of considering Pt-substituted electrocatalysts for other industrial-grade reactors.

## Methods
### Materials

Chloride hexahydrate (NiCl$_2$·6H$_2$O), Boric acid (H$_3$BO$_3$), ammonium chloride (NH$_4$Cl), potassium phosphate dibasic anhydrous (K$_2$HPO$_4$), Potassium dihydrogen phosphate (KH$_2$PO$_4$), Potassium hydroxide (KOH) and polyvinylpyrrolidone (PVP), 25% ammonia (NH$_3$·H$_2$O) were bought from Aladdin, and concentrated sulfuric acid (H$_2$SO$_4$), copper sulfate pentahydrate (CuSO$_4$·5H$_2$O), concentrated nitric acid (HNO$_3$), ethylenediaminetetraacetic acid, potassium thiocyanate acid (KSCN) and ultrapure water (>18.2 MΩ) were purchased from China National Medicines Corporation Ltd. Ruthenium chloride hydrate (RuCl$_3$·xH$_2$O) was obtained from Beijing Innochem Technology Corporation Ltd. All the chemicals were used as received without any other purification.

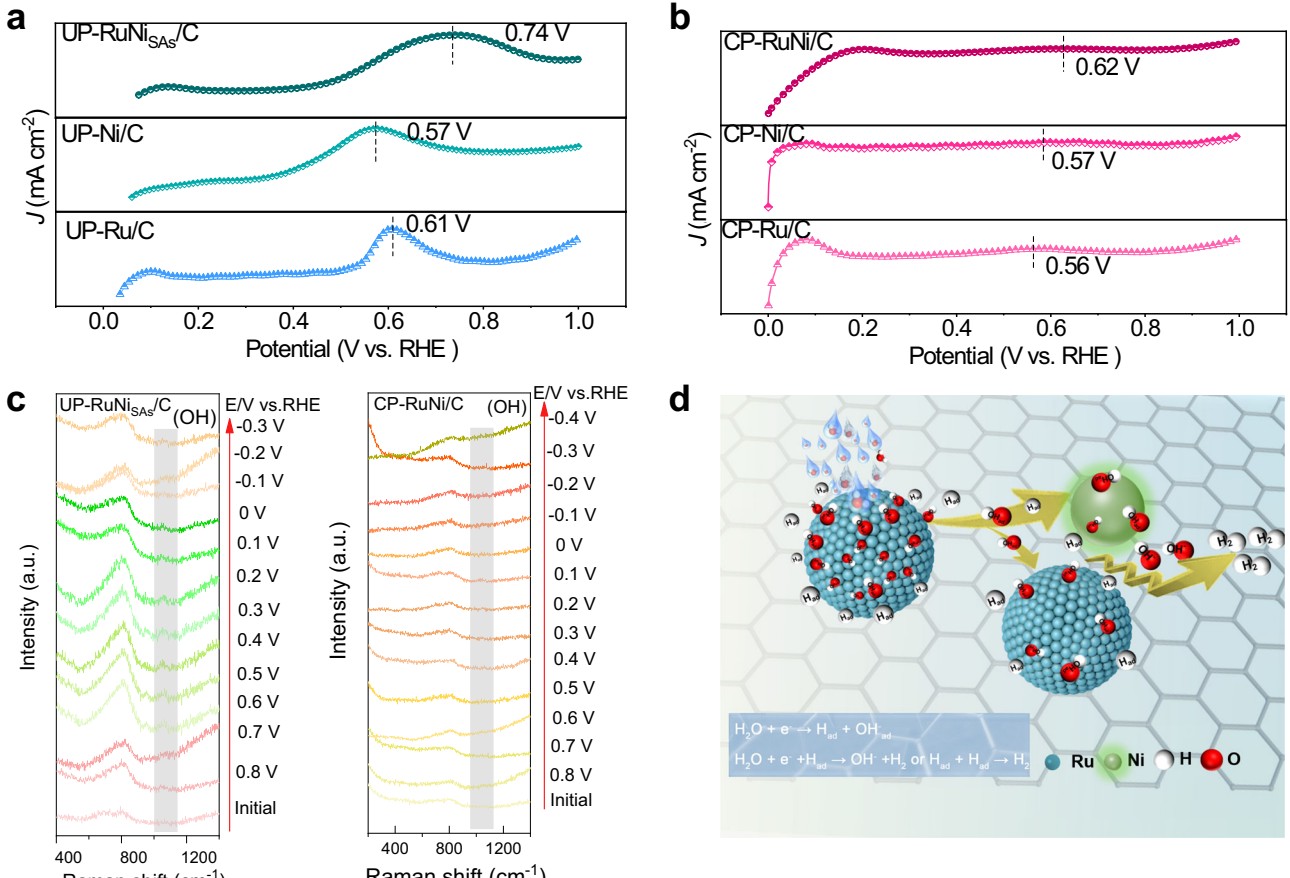

**Fig. 7 | CO-stripping experiment and operando spectroscopic characterizations. a, b** CO-stripping experiment for catalyst UP-Ni/C, UP-Ru/C, and UP-RuNi$_{SAs}$/C obtained by UPED and CP-Ni/C, CP-Ru/C, and CP-RuNi/C obtained by CP. **c** Operando Raman spectroscopy experiments for UP-RuNi$_{SAs}$/C and CP-RuNi/C. **d** Reaction mechanism for the UP-RuNi$_{SAs}$/C catalyst.

## Syntheses

**Synthesis of UP-RuNi$_{SAs}$/C.** A three-electrode attached to an ECLab electrochemical workstation set-up was used to deposit electrocatalysts, with a saturated Ag|AgCl as reference electrode, a platinum wire as counter electrode, and CFP electrode (1 cm *1 cm) as working electrode. Before electrodeposition, the clean carbon paper undergoes a pre-activation process: the clean carbon paper was firstly soaked in 1.0 M $HNO_3$ for 30 min, and then activated by CV in 0.5 M $H_2SO_4$ electrolyte solution for five cycles, of which the scanning range was 0.2–1.5 V vs. RHE. To synthesize UP-RuNi$_{SAs}$/C electrocatalyst, 100 mL of plating solution was prepared, which contains $NiCl_2 \cdot 6H_2O$ (150 mM), $RuCl_3 \cdot xH_2O$ (5 mM), $H_3BO_3$ (160 mM), $NH_4Cl$ (20 mM), PVP (1.0 g). The pH was adjusted to 2.7 using the $NH_3 \cdot H_2O$, and the UP-RuNi$_{SAs}$/C was obtained by UPED method with pulse voltage of −0.8 V (vs. Ag|AgCl) for 4000 times of pulses (8000 s). The catalyst loading mass was decided to be 1.25 mg cm$^{-2}$.

**Synthesis of UP-Ru/C and UP-Ni/C.** The UP-Ru/C, UP-Ni/C electrocatalysts were synthesized by the same method with that of UP-RuNi$_{SAs}$/C, except that without adding $NiCl_2 \cdot 6H_2O$ and $RuCl_3 \cdot xH_2O$ to the plating solution respectively. The catalyst loading mass was decided to be 1.25 mg cm$^{-2}$.

**Synthesis of UP-RuNi$_{SAs}$/Ti.** The UP-RuNi$_{SAs}$/Ti was synthesized by the same method with that of UP-RuNi$_{SAs}$/C, except that the fresh titanium felt as working electrode.

**Synthesis of CP-Ni/C, CP-Ru/C and CP-RuNi/C.** In a typical preparation of CP-RuNi/C, the plating solution with the same composition with that of synthesizing UP-RuNi$_{SAs}$/C was prepared, and the electrocatalyst was obtained by the method of chronopotentiometry plating (CP) with constant voltage of −0.8 V (vs. Ag|AgCl) for 8000 s, and the same method was employed to prepare the CP-Ru/C and CP-Ni/C, except that without adding $NiCl_2 \cdot 6H_2O$ and $RuCl_3 \cdot xH_2O$ to the plating solution respectively. The catalyst loading mass was decided to be 1.25 mg cm$^{-2}$.

**Synthesis of Pt/C catalyst.** To prepare the Pt/C electrocatalyst, 20% commercial Pt/C powder was loaded on the CFP by the method of dripping with the loading mass of 1.25 mg cm$^{-2}$.

## Characterization

The morphology of the catalysts was analyzed using scanning electron microscopy (SEM, 3 kV, Hitachi-SU8010), transmission electron microscopy (TEM, 200 kV, JEOL JEM 2100F) and atomic resolution aberration-corrected HAADF-STEM (FEI Titan G2 60-300). Energy dispersive X-ray spectrometry data were collected on a IXRF SDD 2610 system. XRD was carried out using a Bruker D8 Advance diffractometer with a Cu Kα radiation. ICP-OES was operated using an Agilent 5110. X-ray photoelectron spectroscopy (XPS, Thermo VG ESCALAB250) was used for analyzing surface binding energy. X-ray Absorption Structure (XAS) measurements were performed at the 4B9A beamline of the Beijing Synchrotron Radiation Facility under ring conditions of 2.2 GeV and about 80 mA[43]. The overall energy resolution

of the Ni K-edge is about 1.5 eV due to the spectrometer energy resolution of about 1.4 eV. The EXAFS data processing was carried out using the Athena and Artemis software from the IFEFFIT package[44]. Raman data were collected using the Renishaw inVia.

## Electrochemical measurements

The VMP3 electrochemical workstation was applied to conduct the HER performance of the electrocatalysts in a three-electrode system, in which a graphite rod was applied as the counter electrode, electrocatalyst deposited in situ on CFP as working electrode, and a SCE was used as the reference electrode and the reference was calibrated to the RHE with the calculation formula: $E_{RHE} = E_{SCE} + 0.059*pH + 0.244$. The electrode tests used electrolyte solutions of 1.0 M KOH, 0.5 M $H_2SO_4$, 1.0 M PBS respectively at 25 °C. The LSV test with the scan rate of 1 mV s$^{-1}$ was carried out after 1000 cycles of CV test at 20 mV s$^{-1}$ for activation and investigating the cycling stability. The chronopotentiometry was used to further investigate long-term stability of the electrocatalysts at −10 mA cm$^{-2}$. All the LSV curves were regulated by the 95%-iR compensation. The ECSA was estimated by a CV method, which was carried out at different scan rates v (1, 2, 4, 6, 8, 10 mV s$^{-1}$) with the potential range of 0.1 V, and the calculated formulation is $i_c = v*C_{dl}$, $ECSA = C_{dl}/(C_s*m)$. The electrochemical impedance spectra (EIS) were conducted at approximately −10 mV over the frequency range from 1.0 to $10^5$ Hz.

The TOF value was deliberated by the formula: $TOF = I/(2Fn)$. In which the $I$ represents the corresponding current density at a certain overpotential, $F$ represents the Faraday constant (C mol$^{-1}$), n represents the number of active sites (mol) obtained by the UPD of Cu method, and $n$ is calculated by $n = Q_{Cu}/(2F)$, where $Q_{Cu}$ is the UPD copper stripping charge (Cu upd→Cu$^{2+}$ + 2e$^-$).

To calculate the Faraday Efficiency, we controlled a constant current density to −10 mA cm$^{-2}$, and the $H_2$ produced was quantified by gas chromatographic collection (Agilent, GC-7890B) every 30 min. The relevant calculation formula is as follows:

$$F.E. = \frac{2Fn}{Q} \qquad (1)$$

In which $F$ is the Faraday constant, $n$ is the amount of $H_2$ produced (mol) and the $Q$ is the total charge passed through the cell.

## Co stripping experiment

Co stripping experiment was employed to check the OH adsorption capacity of the catalysts. The working electrode was initially held at a potential of 0.1 V vs RHE in 1.0 M KOH with saturated CO for 10 min. Consequently, the CO in the solution was removed by the $N_2$ flow for 20 min, followed by the CO stripping CV conducted within a potential from 0 to 1.0 V vs RHE, utilizing a scan rate of 5 mV s$^{-1}$.

## AEM electrolyzer

The geometric area of AEM electrolyzer was 1 cm$^2$. The cathode and the anode were sandwiched with the membrane to form AEM electrolyzer, in which the catalyst loading on both cathode and anode are 1.25 mg cm$^{-2}$. The hot 1.0 M KOH solution was flowed through the AEM electrolyzer with a flow velocity of 50 mL min$^{-1}$. The temperature of AEM electrolyzer was measured as 70 °C. The energy efficiencies reported in this work on a lower heating value of $H_2$ basis (LHV) are defined as:

$$\text{Cell efficiency}(\%) = (\Delta H \times n)/(I \times V) \qquad (2)$$

where ΔH is the lower heating value reaction enthalpy for water electrolysis (241.8 kJ/mol). n is the measured hydrogen production rate (mol/s). $I$ is the current applied ($A$). $V$ is the voltage applied ($V$).

## Computational details

To simulate the electronic and energetic properties, we utilized first principles incorporating polarization DFT calculations with the Vienna ab initio simulation program. The calculations were conducted under the generalized gradient approximation with the Perdew–Burke–Ernzerhof formulation. Our study involved the construction of a 21.3 × 12.3 graphene superlayer, which closely resembles the structure of CFP[45,46]. The Ru model is based on the (002) face of a hexagonal crystal. Ni$_{SAs}$ was anchored to the graphene structure by substituted C and substituted carbon defects, respectively. The vacuum spacing perpendicular to the structure plane is 15 Å, and Brillouin zone integration was performed with a 1 × 3 × 1 Monkhorst–Pack k-point mesh. The projected augmented wave potentials were chosen to describe the ionic cores and take valence electrons into account using a plane wave basis set with a kinetic energy cutoff of 450 eV[47,48]. Partial occupancies of the Kohn–Sham orbitals were allowed using the Methfessel-Paxton smearing method and a width of 0.2 eV. The electronic energy was considered self-consistent when the energy change was smaller than 10$^{-5}$ eV and the weak interaction was described by DFT + D3 method using empirical correction in Grimme's scheme[49,50].

## Data availability

The study's supporting data can be found in this article and the Supplementary Information, which are accessible upon reasonable request from the corresponding author.

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

## Acknowledgements

We gratefully acknowledge the financial support from the National Natural Science Foundation of China (U22A20418) (J.-P.L.), (22075196) (G.L.), (21878204) (G.L.), (22250710676) (K.-A.S.) and (22102081) (K.-A.S.), the Key Research and Development Program of Shanxi Province (201903D421073) (G.L.), and Research Project Supported by Shanxi Scholarship Council of China (2022-050) (G.L.).

## Author contributions

R.Y. and J.L. considered and proposed the project. K.S. accomplished the DFT calculations. K.Z., Y.D., and Y.W. contributed to the electrochemical measurement. Q.Z. conducted the structural analysis. G.L., C.C., and Y.S. participated in the discussion of the data and revising the manuscript. R.Y. wrote the manuscript.

## Competing interests

The authors declare no competing interests.
