## [Peer Review file · Nature Communications]

REVIEWER COMMENTS

Reviewer#1

This manuscript reports the Ni single atoms modifying ultra-small Ru nanoparticles with defect carbon bridging structure (UP-RuNiSAs/C) via a unique unipolar pulse electrodeposition (UPED) strategy. It's confirmed that Ru nanocrystals have a high resistance against deactivation because

of the competitive adsorption of OH intermediates on the Ru and single atoms. The experimental results demonstrate the remarkable activity of UP-RuNiSAs/C at

large current densities for HER and the tremendous application foreground. However, there are still some major questions that need to be addressed by the authors. The detailed comments are listed as follows:

1. What are the facets of Ru and Ni used in the DFT calculations? The exposed layer for HER may not be the facets observed from the HRTEM and XRD. It looks more like a few atomic amorphous layers. The model of RuNi/C alloy (CP-RuNi/C) is suggested to be considered to compare with the model of Ni single atom (UP-RuNi/C) in the DFT calculations.
2. From the Fig. 3a, the XRD of UP-RuNi/C and CP-RuNi/C are similar, there are no diffraction peaks corresponding to the Ni. Is the mass of Ni much less than Ru in the RuNi/C alloy for CPRuNi/C catalyst? According to the TEM data, Ru nanocrystals can be observed and have excellent crystallinity, but the XRD data show poor crystallinity and Ru and Ni XRD peaks is too inconspicuous and more likely caused by noise.
3. The authors claimed the spherical aberration correction STEM of Fig. 3e-f. Generally, the resolution of AC-TEM can reach the Angstrom level, and a finer and more accurate structural characterization of the material can be achieved. Moreover, the size of Ni single atom may be angstrom level. Therefore, it's hard to discern the Ru nanocrystals and Ni single atoms from Fig. 3e-f. The AC-TEM with higher resolution is suggested to be provided.
4. The mass content of Ru and Ni should be determined by inductively coupled plasma optical emission spectrometry (ICP-OES).
5. The authors show the SEM and TEM data of CP-RuNi/C, UP-RuNi/C, and UP-Ru/C in the manuscript, but the UP-Ni/C's data? Is it a single atom or nanocrystal Ni in the UP-Ni/C sample?
6. In Fig. 4f, what are the peaks of ~ 853 and 870 eV? The authors should mark it in Fig. 4f.
7. The typical LSV curves without iR-compensated should be provided for comparison in the manuscript.
8. The authors explain that Ni single atoms can be anchored by the carbon defect and show the C-Ni bond model in the DFT calculation. However, to demonstrate the uniqueness of carbon substrate, the authors deposit the same UP-RuNiSAs on titanium felt using the UPED strategy, can Ni single atoms still be anchored without carbon defects? What data proves the existence of Ni single atom in UP-RuNiSAs/Ti? What is the bond between Ni and Ti?

9. Ag/AgCl electrodes are not stable for long-term stability tests in an alkaline solution, a graphite electrode, instead of Pt, should be used as the counter electrode to evaluate the HER performance. Pt will be dissolved under electrochemical conditions and deposited on the cathode, contributing to the HER.

10. Ni is not very stable in alkaline solution; please provide the XPS elemental ratio of UPRuNi/C after the long-term stability test.

Reviewer #2 (Remarks to the Author):

In this paper, the authors prepared RuNiSAs/C as a alkaline HER catalyst with optimal OHad energy. The synthesized RuNiSAs exhibited an overpotential of 9 mV at 10 mA/cm² under 1 M KOH conditions during HER test and maintained a voltage of 1.95 V at 1 A/cm² for 250 h when applied to AEMWE. However, materials characterization seems to be greatly improved. The comments are listed in below to enhance the paper quality.

1. The 2 μm-sized agglomerated particles in Fig 1b are not matched to the ~50 nm-sized particles shown in the magnified TEM image of Fig 1c. The authors should clarify the particle characterization. The line 134, “The rough surface of UP-RuNiSAs/C electrocatalyst agglomerated by ultra-small nanocrystals was proved (Fig. 3b).” is not correct. The particle in Fig 1b is composed of UP-RuNiSAs and “carbon support”. So, its rough surface comes from the carbon support, not the RuNiSAs aggregates. In fig 3c, the authors can discuss “rough surface” of RuNiSAs aggregates. Do not match Fig 1b and 1c. They did not show the same particle.

2. The line 176 discussed “Notably, the CN of Ni-Ru (1.0) makes it reasonable to think that there is a weak electronic interaction between them.”. However, Ni-Ru peak was not assigned in Fig 4b. In addition, Fig 4e also did not show Ni-Ru peak. The authors should define the real structure of RuNiSAs in a clearer way.

3. In a similar way, the line 186, “the Ru K-edge revealed that the average coordination number of Ru in RuNiSAs/C is slightly decreased compared with that in Ru foil, probably because the partial-location environment being occupied by other types of atoms” includes scientific error. In general, Ru nanoparticle has lower CN than the Ru foil due to the nanoparticle effect. So, it can not demonstrate the existence of Ru-Ni.

4. In AEMWE, the performance of the benchmark Pt/C is too bad. What is the performance of commercial Ru/C in AEMWE?

5. Fig 7d is not an appropriate figure format for the manuscript. It can be used as a TOC.

Reviewer #3 (Remarks to the Author):

In this work, the authors fabricated the Ni single atom modified ultra-small Ru nanoparticle structure (UP-RuNiSAs/C) for hydrogen production. The role of Ni atom and defect carbon has been declared by the experiments and DFT simulations. However, there are some flaws on the computational models and analysis, therefore, it cannot be accepted for publication in this journal at the current form.

(1) For the graphene edge there are two typical structures, armchair and zigzag, while in this study, the authors only chose armchair-like graphene as the model. Similarly, for Ni single atom anchoring on the edge, there are also some possible structures, why the authors selected Figure S1d as the model? Is it stable? If the different models were selected, the results may be changed.

(2) "Notably, Nidef/C has the highest $|EOH^*|$ value of 0.32 eV when compared with Ni (111) and Nisub/C models, indicating that it participates the most in the collaborative adsorption of OH^* , thus leading to the smallest $|EOH^*|$ value of 0.15 eV for Ru (002) in RuNidef/C to exposure of active sites." This conclusion is very strange, the strongest adsorption on Nidef/C leading to the weakest adsorption on complex RuNidef/C? The change in the adsorption energy depends on the electronic structure of substrates. Therefore, differential charge density, density of states, and other electronic information for pristine RuNidef/C were required to be calculated to further reveal the essential reasons for the strongest and weakest OH adsorption, as well as the catalytic activity.

(3) Fig. 1c is completely confused, it is better not to rotate the substrate. Also, to distinguish the H atom in H₂O and H atom on the edge, it is recommended to change the color of H in H₂O, similar revision should be done for Fig. S4.

(4) Fig. 1d shows that RuNidef/C possesses a lowest barrier of 0.39 eV, however, it is the endothermic process (~ 0.2 eV), indicating that it is thermodynamically forbidden. Thus, it is difficult to conclude that RuNidef/C is more favorite for water dissociation compared to RuNisub/C and Ru(002).

(5) The authors noted that "Ru nanocrystals have a high resistance against deactivation because of competitive adsorption of OH intermediates (OH_{ad}) on the Ru and single atoms", while on RuNidef/C surface only OH adsorbed on Ni single atom has been shown in the current version, for comparison, the other OH adsorption structures or OH and H co-adsorption structures, such as on C or Ru or Ni-Ru bridge, should be shown in supporting information, as well as the adsorption energy.

Responses to Reviewers

We thank the reviewers for their valuable comments on our manuscript (NCOMMS-23-43720-T). We have carefully considered the reviewers' comments and revised the manuscript accordingly. In this response letter, the reviewers' comments are listed below, with their specific concerns numbered sequentially. Our response is highlighted in blue text, and the revised manuscript and corresponding changes in additional information are highlighted in yellow.

Following is our point-by-point response to all the comments:

Reviewer#1

This manuscript reports the Ni single atoms modifying ultra-small Ru nanoparticles with defect carbon bridging structure (UP-RuNi_{SA}s/C) via a unique unipolar pulse electrodeposition (UPED) strategy. It's confirmed that Ru nanocrystals have a high resistance against deactivation because of the competitive adsorption of OH intermediates on the Ru and single atoms. The experimental results demonstrate the remarkable activity of UP-RuNi_{SA}s/C at large current densities for HER and the tremendous application foreground. However, there are still some major questions that need to be addressed by the authors. The detailed comments are listed as follows:

Response: We appreciate the reviewer for her/his constructive comments and suggestions to enhance the quality of our manuscript. Below, we present point-to-point responses to the reviewers' questions. In this new version of the manuscript, we have addressed all the experimental and theoretical issues raised by the reviewer and made supplementary updates to address these issues by adding corresponding characterization analysis.

1. What are the facets of Ru and Ni used in the DFT calculations? The exposed layer for HER may not be the facets observed from the HRTEM and XRD. It looks more like

a few atomic amorphous layers. The model of RuNi/C alloy (CP-RuNi/C) is suggested to be considered to compare with the model of Ni single atom (UP-RuNi/C) in the DFT calculations.

Response: We greatly appreciate your professional review of our manuscript.

(a) I'm sorry for the lack of rigor in our previous manuscript. For the facets of Ru in the in the DFT calculations, we have placed emphasis on the (002) crystal plane for Ru in the revised manuscript. Additionally, we have thoroughly analyzed and checked the XRD and HRTEM characterization mentioned in the previous manuscript. In order to further reinforce our assertion and theoretical calculations, we employed higher-resolution AC-TEM for UP-RuNi_{SA}s/C.

The following part has been added in the revised manuscript and revised Supplementary information:

Fig. 3 | Physical characterization and morphology of UP-RuNi_{SA}s/C. **e** and **f**, Spherical aberration correction STEM, in which the bright spots highlighted by pink circles are ascribed to Ni single atoms. **g**, STEM-EDS mapping of UP-RuNi_{SA}s/C.

(Please see Fig. 3e-g in the revised manuscript.)

“Fig. 3e revealed the uniform dispersion of bright spots, indicating the uniform distribution of high-contrast single atoms on carbon fiber paper (CFP). Additionally, higher-resolution AC-TEM enable the observation of the scattered bright spots, while

higher brightness Ru crystals can be seen due to the larger atomic number and the Ru (002) surface can be exposed (Fig. 3f). We can preliminarily speculate that the scattered bright spots around the Ru crystal are Ni_{SAs}, which was further validated through STEM-EDS. From Fig. 3g, it can be observed that Ru element is uniformly dispersed at the position of the nanocrystal, while Ni element is uniformly dispersed around the substrate and Ru.”

(Please see Line 10-17 of Page 10 in the revised manuscript.)

“To model the structure of Ru nanocrystals, we focused on the Ru (002) surface of the hexagonal Ru structure (Supplementary Fig. 1a)”

(Please see Line 17-18 of Page 4 in the revised manuscript.)

Supplementary Fig. 21 | Slow scanning XRD patterns of UP-RuNi_{SAs}/C.

Supplementary Note 16 | Slow scanning XRD patterns of the UP-RuNi_{SAs}/C catalyst only detected the partial diffraction peak of the Ru crystal.

(Please see Fig. 21 in the revised Supplementary information.)

Supplementary Fig. 24 | a,b, Spherical aberration correction STEM of UP-RuNiSAs/C.

Supplementary Note 17 | AC-TEM revealed distinct modes of uniform Ru nanocrystal dispersion in UP-RuNiSAs/C. Additionally, the high-resolution STEM images displayed the (002) and (101) crystal planes of Ru nanocrystals, further demonstrating its crystal properties of UP-RuNiSAs/C.

(Please see Fig. 24 in the revised Supplementary information.)

(b) Based on your professional and critical thinking and suggestions, we have carefully added and described the theoretical calculations of RuNi alloy and CP-RuNi/C in the revised manuscript and supporting information used for comparison with UP-RuNiSAs/C.

The following part has been added in the revised manuscript and revised Supplementary information:

Supplementary Fig. 13 | **a**, Initial state, **b**, transition state and **c**, final state of HER process for RuNi alloy model under alkaline conditions.

Supplementary Note 9 | In order to compare with the RuNi_{def}/C model, the RuNi alloy with Ni partially replacing Ru crystals (002) was modeled and its water decomposition energy barrier was calculated.

Supplementary Fig. 14 | Reaction coordinates for HER process for RuNi alloy.

Supplementary Note 10 | The RuNi alloy has a high transition state energy barrier 0.7 eV, even higher than 0.52 eV of pure Ru (002).

Supplementary Fig. 15 | Adsorption of H intermediates for RuNi alloy and RuNi/C with Ni replaces carbon edge.

Supplementary Note 11 | The H^* adsorption energies (ΔG_{H^*}) of the RuNi alloy and RuNi/C are 0.43 eV and 0.34 eV, respectively, indicating that the interaction of RuNi alloys with Ni_{def}/C optimizes the ΔG_{H^*} on the alloy.

(Please see Fig. 13-15 in the revised Supplementary information.)

“In addition, the RuNi alloy model has a higher transition state energy barrier of 0.7 eV (Supplementary Fig. 13-15), which does not reflect an advantage over pure Ru. However, it is worth noting that both RuNi alloy and RuNi_{def}/C optimize H^* adsorption to some extent (Supplementary Fig. 15)”

(Please see Line 17-20 of Page 7 in the revised manuscript.)

2. From the Fig. 3a, the XRD of UP-RuNi/C and CP-RuNi/C are similar, there are no diffraction peaks corresponding to the Ni. Is the mass of Ni much less than Ru in the RuNi/C alloy for CP-RuNi/C catalyst? According to the TEM data, Ru nanocrystals can be observed and have excellent crystallinity, but the XRD data show poor crystallinity and Ru and Ni XRD peaks is too inconspicuous and more likely caused by noise.

Response: Thank you for your thorough review and for alerting us to the oversight. We are sorry for the inaccuracies in characterizing and describing the experimental results

in the previous manuscript. To improve the experimental accuracy and rigor of our study, we have made several amendments based on your comment. First, we carefully supplemented the slow scanning XRD of the CP-RuNi/C catalyst in order to reduce noise interference. Additionally, we conducted ICP-OES testing of both CP-RuNi/C and UP-RuNi_{SAS}/C to accurately determine the content of Ni and Ru in the catalyst. These adjustments have strengthened our experimental conclusions and enhanced the overall validity of our findings.

The following part has been added in the revised manuscript and revised Supplementary information:

“It is worth noting that the slow scanning XRD patterns of the CP-RuNi/C catalyst (Supplementary Fig. 20) only detected the diffraction peak of the Ru crystal. The corresponding diffraction peak of the Ni crystal was very weak or almost difficult to detected due to the lower content of Ni in the catalyst compared to Ru, which has been proven by ICP-OES (Supplementary Fig. 23)”

(Please see Line 3-7 of Page 11 in the revised manuscript.)

Supplementary Fig. 20 | Slow scanning XRD patterns of CP-RuNi/C.

(Please see Fig. 20 in the revised Supplementary information.)

Supplementary Fig. 23 | ICP-OES analysis of UP-RuNi_{SAs}/C and CP-RuNi/C.

(Please see Fig. 23 in the revised Supplementary information.)

3. The authors claimed the spherical aberration correction STEM of Fig. 3e-f. Generally,

the resolution of AC-TEM can reach the Angstrom level, and a finer and more accurate structural characterization of the material can be achieved. Moreover, the size of Ni single atom may be angstrom level. Therefore, it's hard to discern the Ru nanocrystals and Ni single atoms from Fig. 3e-f. The AC-TEM with higher resolution is suggested to be provided.

Response: Thank you for your rigorous comments to improve the quality of our manuscript. We apologize for the lack of accuracy in characterizing and distinguishing between Ni single atoms and Ru nanocrystals in our previous manuscript. To address this issue and enhance the rigor of the manuscript, we have incorporated your suggestion and employed higher-resolution AC-TEM. In addition, combined with X-ray absorption spectroscopy (XAS), we are now able to accurately differentiate between Ni single atoms and Ru nanocrystals in the revised manuscript.

The following part has been added in the revised manuscript:

Fig. 3 | Physical characterization and morphology of UP-RuNiSAs/C. e and f, Spherical aberration correction STEM, in which the bright spots highlighted by pink circles are ascribed to Ni single atoms. g, STEM-EDS mapping of UP-RuNiSAs/C. (Please see Fig. 3e-g in the revised manuscript.)

“NiSAs and Ru nanocrystals can be accurately distinguished and identified using spherical aberration correction scanning TEM (AC-STEM). Fig. 3e revealed the

uniform dispersion of bright spots, indicating the uniform distribution of high-contrast single atoms on carbon fiber paper (CFP). Additionally, higher-resolution AC-TEM enable the observation of the scattered bright spots, while higher brightness Ru crystals can be seen due to the larger atomic number and the Ru (002) crystal plane can be exposed (Fig. 3f). We can preliminarily speculate that the scattered bright spots around the Ru crystal are Ni_{SAS}, which was further validated through STEM-EDS. From Fig. 3g, it can be observed that Ru element is uniformly dispersed at the position of the nanocrystal, while Ni element is uniformly dispersed around the substrate and Ru.”

(Please see Line 8-17 of Page 10 in the revised manuscript.)

Fig. 4 | Characterizations on chemical states and fine structures. a, Ni K-edge XANES of UP-RuNi_{SAS}/C, NiO, Ni₂O₃ and Ni foil. **b**, Fourier transforms EXAFS spectra at the Ni K-edge of UP-RuNi_{SAS}/C, NiO, Ni₂O₃ and Ni foil. **c**, Wavelet transforms for the k₂-weighted EXAFS signs of UP-RuNi_{SAS}/C, NiO, and Ni foil. **d**, Ni K-edge XANES of UP-RuNi_{SAS}/C, Ru foil and RuO₂. **e**, Fourier transform EXAFS spectra of UP-RuNi_{SAS}/C, Ru foil and RuO₂ at Ru K-edge.

(Please see Fig. 4a-e in the revised manuscript.)

“Compared with the NiO, Ni₂O₃ and Ni-foil, the Ni K-edge X-ray absorption near-edge structure (XANES) spectra for UP-RuNi_{SAS}/C was markedly different with the edge of

the Ni foil and NiO, with the white line between them, confirming its distinctive coordination structure (Fig. 4a).”

(Please see Line 11-13 of Page 12 and Line 1 of Page 13 in the revised manuscript.)

“As exhibited in Fig. 4b for the Fourier transformed (FT) k_2 -weighted extended X-ray absorption fine structure (EXAFS) spectrum of UP-RuNi₅As/C, no metal bonding peaks were found at 2-3 Å just as NiO, Ni₂O₃ and Ni-foil, corroborating that the Ni is atomically dispersed in the UP-RuNi₅As/C.”

(Please see Line 5-8 of Page 13 in the revised manuscript.)

“The existence of Ru-Ru coordination for RuNi₅As/C is evident from the comparison with Ru foil, as UP-RuNi₅As/C showed distinct Ru-Ru path at around 2.65 Å by the EXAFS results in Fig. 4e”.

(Please see Line 3-5 of Page 14 in the revised manuscript.)

4. The mass content of Ru and Ni should be determined by inductively coupled plasma optical emission spectrometry (ICP-OES).

Response: We are greatly grateful to the reviewer for your thoughtful consideration. Following up your valuable comments, we have provided inductively coupled plasma optical emission spectrometry (ICP-OES) results of catalysts UP-RuNi₅As/C and CP-RuNi/C in the revised Supplementary information to accurately determine the content of Ru and Ni.

The following part has been added in the revised manuscript and revised Supplementary information:

“ICP-OES accurately analyzed the Ru and Ni contents in the catalysts, revealing atomic ratios of 92.32/7.68 for Ru/Ni in UP-RuNi₅As/C and 81.73/18.27 in CP-RuNi/C respectively (Supplementary Fig. 23)”.

(Please see Line 17-20 of Page 10 in the revised manuscript.)

Supplementary Fig. 23 | ICP-OES analysis of UP-RuNi_{SAs}/C and CP-RuNi/C.

(Please see Fig. 23 in the revised Supplementary information.)

5. The authors show the SEM and TEM data of CP-RuNi/C, UP-RuNi/C, and UP-Ru/C in the manuscript, but the UP-Ni/C's data? Is it a single atom or nanocrystal Ni in the UP-Ni/C sample?

Response: We sincerely thank the reviewer for careful inspection and bringing the oversight to our attention. We apologize for not providing complete data information to support our assertion due to our oversight. To rectify this, we carefully reviewed the manuscript and added data on UP-Ni/C to obtain its structural information more rigorously.

The following part has been added in the revised Supplementary information:

Supplementary Fig. 28 | **a, b**, SEM, **c**, TEM image, **d**, HRTEM image of UP-Ni/C.

Supplementary Note 18 | The results show that the surface of the UP-Ni/C catalyst exhibits the morphology of nanoparticles, and the corresponding nanocrystals exposed the (200) and (111) crystal planes of Ni, proving that Ni in the catalyst is nanocrystals.

(Please see Fig. 28 in the revised Supplementary information.)

6. In Fig. 4f, what are the peaks of ~ 853 and 870 eV? The authors should mark it in Fig. 4f.

Response: We sincerely appreciate the reviewer's thoughtful comment. In response to your suggestion, we have included in the revised manuscript that the binding energies of approximately 853 eV (specifically 852.51 eV) and 870 eV (specifically 869.71 eV),

as shown in Fig. 4f, are linked to the Ni⁰.

The following part and has been added in the revised manuscript:

Fig. 4 | Characterizations on chemical states and fine structures. **f**, High-resolution XPS spectra of Ni 2p for UP-Ni/C and UP-RuNiSAs/C catalysts.

(Please see Fig. 4f in the revised manuscript.)

“In contrast to UP-RuNiSAs/C, the fitted Ni 2p spectrum in UP-Ni/C displayed Ni⁰ peaks at 852.51 eV and 869.71 eV.”

(Please see Line 18-19 of Page 14 in the revised manuscript.)

7. The typical LSV curves without iR-compensated should be provided for comparison in the manuscript.

Response: We are greatly grateful to the reviewer for your professional advice. According to your valuable comments, the typical LSV curves without iR-compensated has been carefully described in the revised manuscript and revised Supplementary information.

The following part has been added in the revised manuscript and revised Supplementary information:

“The LSV curves without iR-compensated were provided in the Supplementary Fig. 37, which is consistent with Fig. 5a that the optimal HER performance of UP-RuNi_{SA}s/C.”

(Please see Line 12-14 of Page 17 in the revised manuscript.)

Supplementary Fig. 37 | Polarization curves without iR-compensated of the UP-Ni/C, UP-Ru/C, CP-RuNi/C, UP-RuNi_{SA}s/C and commercial Pt/C catalysts in 1.0 M KOH.

(Please see Fig. 37 in the revised Supplementary information.)

8. The authors explain that Ni single atoms can be anchored by the carbon defect and show the C-Ni bond model in the DFT calculation. However, to demonstrate the uniqueness of carbon substrate, the authors deposit the same UP-RuNi_{SA}s on titanium felt using the UPED strategy, can Ni single atoms still be anchored without carbon defects? What data proves the existence of Ni single atom in UP-RuNi_{SA}s/Ti? What is the bond between Ni and Ti?

Response: We are truly grateful to the reviewer for these professional advices. We apologize for the lack of adequate evidence in our previous manuscript regarding the presence of Ni and the bonds formed between Ni and Ti in the UP-RuNi₅As/Ti catalyst. To enhance the rigor of our judgment, we performed additional tests on the catalyst's AC-TEM and TEM-EDS mapping to show the structural information and existence form of Ni.

(a) Firstly, we examined the form of Ni present in the UP-RuNi₅As/Ti catalyst by spherical aberration correction STEM.

The following part has been added in the revised manuscript and revised Supplementary information:

Supplementary Fig. 38 | a, SEM, b-c, Spherical aberration correction STEM, in which the bright spots highlighted by pink circles are ascribed to Ni single atoms. d, STEM-EDS mapping of UP-RuNiSAs/Ti.

Supplementary Note 21 | AC-TEM revealed uniform Ru nanocrystal dispersion in UP-RuNiSAs/Ti. Additionally, the high-resolution STEM images displayed the (002) and (100) crystal planes of Ru nanocrystals and dispersed NiSAs.

(Please see Fig. 38 in the revised Supplementary information.)

(b) Regarding your concerns about the bond between Ni and Ti, we validate and explain our conclusions by reviewing recent reports and additional specific characterizations. According to previous research results (*Adv. Mater* 2021, 33, 2102801 and *Nat. Mater.* 2019, 18, 620), metal single atoms such as Ni can be anchored to metal oxide carriers

via Ni-O bridging. By covalently bonding with surface O atoms to form Ni-O bonds, active atomic metals (Ni) can be dispersed and stabilized onto oxide supports, especially for the oxidized state carriers of Ti have been reported. Therefore, based on the pretreatment process (CV oxidation in 0.5 M H₂SO₄) before catalyst deposition and the above conclusions, it is not difficult to speculate that Ni_{SAs} are anchored to Ti via O bridging. Furthermore, we corroborated this speculation by TEM-EDS mapping.

The following part has been added in the revised Supplementary information:

Supplementary Fig. 39 | TEM-EDS mapping of UP-RuNi_{SAs}/Ti.

Supplementary Note 22 | The homogeneous distribution of the four elements and especially the homogeneously dispersed positions of O, Ti and Ni roughly demonstrate the anchoring of Ni_{SAs} on the Ti substrate.

Following up your valuable comments, **the following references have been added in the revised manuscript to focus on the scientific merits and avoid dispute:**

38. Zhao Y, et al. Anchoring Sites Engineering in Single-Atom Catalysts for Highly Efficient Electrochemical Energy Conversion Reactions. *Adv. Mater.* 33, 2102801 (2021).

39. Lee B-H, et al. Reversible and cooperative photoactivation of single-atom Cu/TiO₂ photocatalysts. *Nat. Mater.* 18, 620-626 (2019).

9. Ag/AgCl electrodes are not stable for long-term stability tests in an alkaline solution,

a graphite electrode, instead of Pt, should be used as the counter electrode to evaluate the HER performance. Pt will be dissolved under electrochemical conditions and deposited on the cathode, contributing to the HER.

Response: Thank you for reviewing carefully and bringing the oversight to our attention.

(a) As professionally commented by the reviewer, the Ag/AgCl reference electrode is unstable in alkaline solutions, especially during long-term stability testing. Therefore, in the previous manuscript and Supplementary information, we used the saturated calomel electrode (SCE) as a reference electrode for HER performance testing and corrected the calomel electrode. However, it is precisely due to the kind reminder from the reviewers that we suspect that the calomel electrode will face the same problem.

Based on your valuable reminder, we meticulously reviewed the literature to confirm the accuracy of HER performance testing with the SCE as reference electrode. In 1.0 M KOH solution, we used a Hg/HgO electrode as a reference electrode and tested the HER performance of UP-RuNi₅As/C. The results showed that the polarization curve obtained were almost identical with that of SCE. Furthermore, the overpotential of UP-RuNi₅As/C measured by the Hg/HgO electrode at -10 mA cm⁻² was 10 mV, which was only 1 mV different from the results measured by the SCE. Therefore, considering the allowable range of error, it can be concluded that the test results with the SCE as reference electrode are accurate.

The following part has been added in the revised Supplementary information:

Supplementary Fig. 41 | Polarization curves of UP-RuNi_{SAs}/C catalysts with the SCE and Hg/HgO as reference electrodes respectively in 1.0 M KOH.

Supplementary Note 23 | The results show that the two curves almost coincide. Furthermore, the overpotential of UP-RuNi_{SAs}/C measured by the Hg/HgO electrode at 10 mA cm⁻² was 10 mV, which was only 1 mV different from the results measured by the SCE. Therefore, considering the allowable range of error, it can be concluded that the test results with the SCE as reference electrode are accurate.

(Please see Fig. 41 in the revised Supplementary information.)

(b) We greatly appreciate your suggestion regarding your concerns regarding the use of counter electrodes in performance testing. As you commented, the performance test results using Pt as the counter electrode are not accurate because Pt will dissolve and further deposit onto the cathode. Therefore, in our previous manuscript and Supporting information, we used a graphite electrode as the counter electrode for HER performance testing.

The following part has been added in the revised Supplementary information for the ccatalysts preparation and performance testing:

“Synthesis of UP-RuNi_{SAs}/C, CP-RuNi/C, UP-Ru/C, UP-Ni/C, CP-Ru/C, CP-Ni/C, UP-RuNi_{SAs}/Ti and Pt/C

A three-electrode attached to an EC-Lab electrochemical workstation set-up was used to deposit electrocatalysts, with a saturated Ag/AgCl as reference electrode, a platinum wire as counter electrode and carbon fiber paper (CFP) electrode (1 cm *1 cm) as working electrode.”

(Please see Page 3 in the revised Supplementary information.)

“Electrochemical measurements

The VMP3 electrochemical workstation was applied to conduct the HER performance of the electrocatalysts in a three-electrode system, in which the electrocatalyst deposited in situ on CFP as working electrode, a graphite rod as the counter electrode and a saturated calomel electrode (SCE) as the reference electrode, and the reference was calibrated to the RHE with the calculation formula: $E_{RHE} = E_{SCE} + 0.059 * pH + 0.244$.”

(Please see Page 4 in the revised Supplementary information.)

10. Ni is not very stable in alkaline solution; please provide the XPS elemental ratio of UP-RuNi/C after the long-term stability test.

Response: We sincerely appreciate the reviewer's professional comment. In response to your comment, we have included the XPS element ratios of Ni after the long-term stability test in the revised supporting information. We carefully reviewed the manuscript due to your reminder. In order to further verify the stability of Ni during the long-term testing process, we detected the Ni content in the electrolyte after the long-term stability test using ICP-OES. The corresponding analysis and description have been provided in the revised manuscript.

The following part has been added in the revised Supplementary information:

Supplementary Fig. 51 | **a**, The XPS full spectra, **b**, the high-resolution XPS spectra of C 1s and Ru 3d, **c**, Ni 2p and **d**, Ru 3p of the UP-RuNi₅As/C catalyst after HER test in 1.0 M KOH.

Supplementary Note 24 | After long-term stability testing (100 h), the atomic ratio of each element in the UP-RuNi₅As/C is Ru/Ni/C=5.43 at%/1.43 at%/93.14 at%. After 50 h and 100 h of testing, the electrolyte's ICP-OES analysis detected the Ni content to be -0.048mg/L and -0.045mg/L, respectively, indicating that there was virtually no presence of Ni. This finding provides further evidence of the exceptional stability of Ni in UP-RuNi₅As/C, as it did not dissolve during long-term stability testing.

(Please see Fig. 51 in the revised Supplementary information.)

Reviewer #2 (Remarks to the Author):

In this paper, the authors prepared RuNi₅As/C as an alkaline HER catalyst with optimal OH_{ad} energy. The synthesized RuNi₅As exhibited an overpotential of 9 mV at 10 mA/cm² under 1 M KOH conditions during HER test and maintained a voltage of 1.95 V at 1 A/cm² for 250 h when applied to AEMWE. However, materials characterization seems to be greatly improved. The comments are listed in below to enhance the paper quality.

Response: Thank you for your comments and support of our work. We especially appreciate your detailed comments and valuable suggestions. The comments have been addressed point by point for all the experimental analyses and expressions you suggested and have been changed accordingly in the revised manuscript.

1. The 2 μm-sized agglomerated particles in Fig 1b are not matched to the ~50 nm-sized particles shown in the magnified TEM image of Fig 1c. The authors should clarify the particle characterization. The line 134, “The rough surface of UP-RuNi₅As/C electrocatalyst agglomerated by ultra-small nanocrystals was proved (Fig. 3b).” is not correct. The particle in Fig 1b is composed of UP-RuNi₅As and “carbon support”. So, its rough surface comes from the carbon support, not the RuNi₅As aggregates. In fig 3c, the authors can discuss “rough surface” of RuNi₅As aggregates. Do not match Fig 1b and 1c. They did not show the same particle.

Response: We thank the reviewers for their careful and professional comments, reminding us to critically describe and analyze the characterization results and phenomena. We are sorry for not providing sufficient data to support our assertions in the previous manuscript, which resulted in an incorrect representation of nanoparticle aggregates, nanocrystals and rough surface for UP-RuNi₅As/C. We have added convincing characterizations to the revised manuscript and supporting information, and have modified the corresponding experimental analyses.

The following part has been added in the revised manuscript and revised Supplementary information:

Supplementary Fig. 22 | a, b, SEM image of UP-RuNiSAs/C.

(Please see Fig. 22 in the revised Supplementary information.)

Fig. 3 | Physical characterization and morphology of UP-RuNiSAs/C. b, SEM image. c, TEM image (inset: histogram of size distribution).

(Please see Fig. 3b-c in the revised manuscript.)

“As can be seen from Fig. 3b and Supplementary Fig. 22a, agglomerate-like particles of about 2 μm in size were uniformly loaded on the carbon substrate, and further

magnification (Supplementary Fig. 22b) showed that these particles of about 2 μm in size are agglomerated from smaller nanoparticles to form a rough surface. The TEM image and the corresponding size distribution plot (Fig. 3c) further demonstrated that the size of the nanocrystalline is about 2 nm.”

(Please see Line 2-7 of Page 10 in the revised manuscript.)

2. The line 176 discussed “Notably, the CN of Ni-Ru (1.0) makes it reasonable to think that there is a weak electronic interaction between them.”. However, Ni-Ru peak was not assigned in Fig 4b. In addition, Fig 4e also did not show Ni-Ru peak. The authors should define the real structure of RuNi_{SAs} in a clearer way.

Response: We thank the reviewers for their professional comments reminding us to describe the characterization results critically. We apologize for the lack of a clear and correct representation of the true structure of the UP- $\text{RuNi}_{\text{SAs}}/\text{C}$ catalyst and the interaction between Ru and Ni in the previous manuscript, and we have corrected this error in the revised manuscript.

The following part has been added in the revised manuscript:

“As exhibited in Fig. 4b for the Fourier transformed (FT) k^2 -weighted extended X-ray absorption fine structure (EXAFS) spectrum of UP- $\text{RuNi}_{\text{SAs}}/\text{C}$, no metal bonding peaks were found at 2-3 \AA just as NiO, Ni_2O_3 and Ni-foil, corroborating that the Ni is atomically dispersed in the UP- $\text{RuNi}_{\text{SAs}}/\text{C}$ ”.

(Please see Line 5-8 of Page 13 in the revised manuscript.)

“For UP- $\text{RuNi}_{\text{SAs}}/\text{C}$, no Ni-Ni WT signal was observed in Fig. 4c unlike the Ni foil and NiO at $\sim 8.2 \text{\AA}^{-1}$. We speculated that it may be the coordination of Ni with C at $\sim 6.0 \text{\AA}^{-1}$. We verified the above supposition using the EXAFS fitting parameters at the Ni K-edge listed in Supplementary Fig. 29 and Supplementary Table 5. As expected, the coordination structure of Ni-C obtained by fitting the Ni_{SAs} path is the optimal one and

the coordination number (CN) is much larger than that of Ni-O (which could be ignored). It is worth noting that, as inferred from the absence of Ni-Ru bond in Fig. 4b and the CN of Ni/Ru value (1.0) in Supplementary Table 5, there is no bonding between Ni and Ru, and there may be weak electronic interactions. The above results confirm the successful synthesis of Ni_{SAs} anchored carbon structures.”

(Please see Line 11-20 of Page 13 in the revised manuscript.)

3. In a similar way, the line 186, “the Ru K-edge revealed that the average coordination number of Ru in UP-RuNi_{SAs}/C is slightly decreased compared with that in Ru foil, probably because the partial-location environment being occupied by other types of atoms” includes scientific error. In general, Ru nanoparticle has lower CN than the Ru foil due to the nanoparticle effect. So, it can not demonstrate the existence of Ru-Ni.

Response: Thank you for reviewing carefully and bringing the oversight to our attention the scientific errors in the previous description. We apologize for the lack of professionalism in our previous manuscript, which led to the misrepresentation of the coordination number concept and the interaction between Ru and Ni. Based on the reviewer's reminder, we have carefully examined the manuscript and have corrected these errors in the revised manuscript.

The following part has been added in the revised manuscript:

“The EXAFS fitting parameters in Supplementary Fig. 31 and Supplementary Table 6 at the Ru K-edge revealed that the slight decrease in the average coordination number of Ru in UP-RuNi_{SAs}/C compared to Ru foil, possibly due to the smaller size of Ru nanoparticles in the UP-RuNi_{SAs}/C catalysts (nano effect), and the increase in the number of surface atoms ultimately leads to insufficient number of coordinating atoms. It is preliminarily concluded that Ni_{SAs}-anchoring carbon modifying Ru nanocrystalline materials have been successfully synthesized.”

(Please see Line 7-13 of Page 14 in the revised manuscript.)

4. In AEMWE, the performance of the benchmark Pt/C is too bad. What is the performance of commercial Ru/C in AEMWE?

Response: We sincerely appreciate the reviewer's professional advice. Based on the reviewers' comments, we scrutinized the previous manuscript and additionally tested the performance and stability of commercial Ru/C catalysts as cathodes in AEMWE.

The following part has been added in the revised manuscript and revised Supplementary information:

Supplementary Fig. 57 | LSV curves of the AEM reactors using NiFeO_x as the anodic and UP-RuNiSAs/C, commercial Pt/C and Ru/C as the cathodic electrodes, respectively at 70 °C

Supplementary Fig. 58 | Stability tests of the AEM water electrolyzers at 1 A cm^{-2} using NiFeO_x as the anodic and UP-RuNiSAs/C , commercial Pt/C and Ru/C as the cathodic electrodes, respectively at $70 \text{ }^\circ\text{C}$

(Please see Fig. 57-58 in the revised Supplementary information.)

“We used the commercial NiFeO_x catalyst as the anodic and compared the performance of UP-RuNiSAs/C with the commercial Pt/C and Ru/C catalysts as the cathodic separately”

(Please see Line 12-13 of Page 20 in the revised manuscript.)

“Surprisingly, low cell voltage of 1.70 V and 1.95 V were recorded at 0.5 A cm^{-2} and 1 A cm^{-2} respectively at $70 \text{ }^\circ\text{C}$ for UP-RuNiSAs/C catalyst, whereas the commercial Pt/C and Ru/C catalysts obtained a cell voltage of 2.27 V and 1.91 V separately at 500 mA cm^{-2} (Fig. 6c and Supplementary Fig. 57)”.

(Please see Line 1-4 of Page 21 in the revised manuscript.)

“As shown in Fig. 6e and Supplementary Fig. 58, the UP-RuNiSAs/C gave a great long-term stability with negligible potential fluctuations compared with the Pt/C and Ru/C catalysts, suggesting the promising prospect of UP-RuNiSAs/C for hydrogen production.”

(Please see Line 8-11 of Page 21 in the revised manuscript.)

5. Fig 7d is not an appropriate figure format for the manuscript. It can be used as a TOC.

Response: We are completely grateful to the reviewer for your thoughtful contemplation. We apologize that we did not provide the proper graphic formatting for Fig 7d in the previous manuscript, and after careful examination and revision, we have corrected this error in the revised manuscript.

The following part has been added in the revised manuscript:

Fig. 7 | CO-stripping experiment and operando spectroscopic characterizations. d, Reaction mechanism for the UP-RuNiSAs/C catalyst.

(Please see Fig. 7d in the revised manuscript)

Reviewer #3 (Remarks to the Author):

In this work, the authors fabricated the Ni single atom modified ultra-small Ru nanoparticle structure (UP-RuNi₁SA_s/C) for hydrogen production. The role of Ni atom and defect carbon has been declared by the experiments and DFT simulations. However, there are some flaws on the computational models and analysis, therefore, it cannot be accepted for publication in this journal at the current form.

Response: Thank you for your valuable comments on our work. We greatly appreciate your efforts and valuable comments on our work. Based on your valuable comments, we have made detailed additions. We have further dealt with the comments point by point for some deficiencies in the computational model and analysis, and have revised the revised manuscript and Supporting information accordingly.

(1) For the graphene edge there are two typical structures, armchair and zigzag, while in this study, the authors only chose armchair-like graphene as the model. Similarly, for Ni single atom anchoring on the edge, there are also some possible structures, why the authors selected Figure S1d as the model? Is it stable? If the different models were selected, the results may be changed.

Response: We feel great thanks for your professional review work on our manuscript. Due to our incomplete consideration, we did not list the complete information about the graphene edge and the anchoring mode of Ni₁SA_s at the edge in the previous manuscript. Based on the reviewer's professional comments for improving the quality of our work, we have compared the formation energy, adsorption energy, and water decomposition energy barriers of the two graphene edge models.

The following part has been added in the revised manuscript and revised Supplementary information:

Supplementary Fig. 3 | Theoretical calculation models of RuNi_{def}/C with two typical structures for the graphene edge. a, armchair models, b, zigzag models.

Supplementary Note 1 | The formation energies of the RuNi_{def}/C with armchair-like and zigzag-like graphene edge are both -3.36 eV, indicating that the two models are equally stable.

Supplementary Fig. 4 | Adsorption of OH and H intermediates for RuNi_{def}/C with **a, b** armchair-like and **c, d** zigzag-like graphene edge.

Supplementary Note 2 | The OH* adsorption energies (E_{OH^*}) of the RuNi_{def}/C with armchair-like graphene edge (A-RuNi_{def}/C) and zigzag-like graphene edge (Z-RuNi_{def}/C) are 0.39 eV and 0.42 eV, respectively, indicating that both have similar adsorption capacities for OH*, whereas the ΔG_{H^*} of the corresponding two models are -0.18 and -0.15, indicating that zigzag-like graphene edge has an advantage from this point of view.

Supplementary Fig. 5 | Partial reaction coordinates for HER process for A-RuNi_{def}/C and Z-RuNi_{def}/C in order to screen graphene edge types.

Supplementary Note 3 | Apparently, RuNi_{def}/C with zigzag-like graphene edge has a lower energy barrier for the water decomposition transition state, so the RuNi_{def}/C with zigzag-like graphene edge was used for a series of subsequent studies. It is noteworthy that the energy barrier for water decomposition transition state, OH* adsorption, and ΔG_H* of RuNi_{def}/C with both armchair-like and zigzag-like graphene edge are superior to those of pure Ni_{def} and Ru (002).

(Please see Fig. 3-5 in the revised Supplementary information.)

“It is important to note that we performed an initial screening of the substrate carbon model in terms of formation energy, intermediate adsorption, and water decomposition barriers, respectively (Supplementary Fig. 3-8). We ultimately determined that the most stable and suitable Ni_{SAs}-anchored carbon defect is the zigzag-like graphene edge (Supplementary Fig. 3b).”

(Please see Line 21 of Page 4 and Line 1-4 of Page 5 in the revised manuscript.)

(2) “Notably, Ni_{def}/C has the highest |E_{OH*}| value of 0.32 eV when compared with Ni

(111) and Ni_{sub}/C models, indicating that it participates the most in the collaborative adsorption of OH*, thus leading to the smallest |E_{OH*}| value of 0.15 eV for Ru (002) in RuNi_{def}/C to exposure of active sites.” This conclusion is very strange, the strongest adsorption on Ni_{def}/C leading to the weakest adsorption on complex RuNi_{def}/C? The change in the adsorption energy depends on the electronic structure of substrates. Therefore, differential charge density, density of states, and other electronic information for pristine RuNi_{def}/C were required to be calculated to further reveal the essential reasons for the strongest and weakest OH adsorption, as well as the catalytic activity.

Response: We are very grateful to the reviewer for his/her professional and critical comments.

(a) We apologize that in the previous manuscript, we did not carefully review the description related to OH adsorption at a particular site, which in turn caused some misrepresentation. Therefore, in the revised manuscript, we have explained and corrected these errors in detail to make our presentation clearer. In addition, our experimental results (CO-stripping experiments) are also in agreement with this theoretical calculation result.

The following part has been added in the revised manuscript and revised Supplementary information:

Fig. 1 | Theoretical prediction results by DFT calculation. a, Adsorption energy of OH intermediate on active sites for HER corresponding to different models.

(Please see Fig. 1a in the revised manuscript)

Supplementary Table 1 | Adsorption energy of various intermediate adsorption on active sites for HER corresponding to different models.

Model	E_{OH^*} (eV)	$ E_{H^*} $ (eV)
Ru (002)	-0.43	0.53
Ni (111)	-0.25	0.71
Ni _{sub}	0.29	0.72
Ni _{def}	0.32	0.55
RuNi _{sub} /C	0.35	0.24
RuNi _{def} /C	-0.15	0.15

Fig. 7 | CO-stripping experiment and operando spectroscopic characterizations. a,
CO-stripping experiment for catalyst UP-Ru/C and UP-RuNiSAs/C.

(Please see Fig. 7a in the revised manuscript)

“The adsorption of OH* was precisely investigated firstly through the calculating the adsorption energy (E_{OH^*}) for different models in Fig. 1a and Supplementary Table 1. Notably, Ni_{def}/C has the weakest adsorption of OH* ($E_{OH^*}=0.32$ eV) compared to the Ni (111) of -0.25 and Ni_{sub}/C of 0.29, and Ni_{def}/C has a superior free energy of H* ($|E_{H^*}|$) compared to Ni_{sub}/C, suggesting that NiSAs is more favorable for weakening OH* adsorption on Ru sites relative to metal Ni, resulting in the optimal OH* adsorption energy for Ru (002) in RuNi_{def}/C ($E_{OH^*}=-0.15$ eV) for exposure of active sites”

(Please see Line 4-10 of Page 5 in the revised manuscript.)

“Electrochemical oxidation of CO is triggered by adsorption of active OH, and thus the negative shift of the oxidation potential is generally considered indicative of a stronger OH affinity for the catalyst. As shown in Fig. 7a, the UP-RuNiSAs/C demonstrated the lowest OH adsorption capacity with the peak potentials of 0.74 V compared with the pure UP-Ru/C (0.61 V), implying the better OH desorption kinetics of Ru site in UP-RuNiSAs/C with the introduction of NiSAs during the Volmer step”

(Please see Line 17-22 of Page 21 in the revised manuscript.)

(b) To address your concerns about the correlation between OH adsorption energies

and electronic structure, we recalculated the OH adsorption energies and projected density of states (PDOS) for other models associated with the RuNi_{def}/C (i.e. Z-RuNi_{def}/C) model to validate the changes in OH adsorption energies caused by changes in the electronic structure of RuNi_{def}/C.

The following part has been added in the revised manuscript and revised Supplementary information:

Supplementary Fig. 6 | Adsorption of OH intermediates on the Ni site for **a**, Z-Ni_{def}/C and **b**, Z-RuNi_{def}/C.

Supplementary Note 4 | The OH* adsorption energies (E_{OH^*}) on the Ni site for Z-Ni_{def}/C and Z-RuNi_{def}/C are 0.32 eV and 0.39 eV, respectively, indicating that the introduction of Ru site weakened the adsorption of OH at the Ni site due to the interaction between Ru and Ni.

Fig. 1 | Theoretical prediction results by DFT calculation. c, PDOS of Ni for $\text{Ni}_{\text{def}}/\text{C}$ ($\text{Z-Ni}_{\text{def}}/\text{C}$) and $\text{RuNi}_{\text{def}}/\text{C}$ ($\text{Z-RuNi}_{\text{def}}/\text{C}$).

Supplementary Fig. 7 | OH intermediates are adsorbed on the Ru site with the adsorbed Ru located at different distances from Ni: **a**, Ru (002), **b**, near position (RuNi_{def}/C-N i.e. RuNi_{def}/C), **c**, middle position (RuNi_{def}/C-M) and **d**, far position (RuNi_{def}/C-F).

Supplementary Note 5 | The OH* adsorption energies (E_{OH^*}) on the Ru site for Ru (002), RuNi_{def}/C-N, RuNi_{def}/C-M and RuNi_{def}/C-F are -0.43 eV, -0.15 eV, -0.18 eV and -0.23 eV, respectively, indicating that the adsorption of OH at the Ru site is weakened due to the interaction between Ru and Ni, and the closer the distance between Ru and Ni, the stronger this weakening effect is. It is worth noting that the best model in this manuscript is RuNi_{def}/C-N.

Fig. 1 | Theoretical prediction results by DFT calculation. d, PDOS of Ru for Ru(002) and RuNi_{def}/C.

Supplementary Fig. 3 | Theoretical calculation models of RuNi_{def}/C with two typical structures for the graphene edge. a, armchair models, b, zigzag models.

Supplementary Fig. 9 | PDOS of Ni for Ni_{def}/C and RuNi_{def}/C with armchair-like (A-RuNi_{def}/C) and zigzag-like graphene edge (Z-RuNi_{def}/C).

Supplementary Note 7 | The d-bands centers of the two models (A-RuNi_{def}/C and Z-RuNi_{def}/C) are very close.

“In addition, the partial density of states (PDOS) was used to examine the electronic interactions between the Ru and Ni sites and thus to analyze the causes of the OH adsorption energy changes. As shown in Fig. 1c, the introduction of Ni_{def} into Ru caused a downward shift of the d-band center (-1.27 eV to -2.26 eV), and therefore caused the weakening of OH adsorption on the Ni site. While as the Ni_{def} enters the pure Ru system, the d-band center shifts downward from -1.63 eV to -2.86 eV (Fig. 1d) due to the rearrangement of electrons around Ru, and the adsorption of Ru on OH is weakened. This result is consistent with the calculation of OH adsorption energy. Notably, the electronic interactions between Ru and Ni weaken the adsorption of OH by RuNi_{def}/C, thus reducing the final water decomposition energy barrier.”

(Please see Line 18-22 of Page 5 and Line 1-5 of Page 6 in the revised manuscript.)

(3) Fig. 1c is completely confused, it is better not to rotate the substrate. Also, to distinguish the H atom in H₂O and H atom on the edge, it is recommended to change the color of H in H₂O, similar revision should be done for Fig. S4.

Response: Thank you for careful reviewing and bringing the oversight to our attention. We apologize that in the previous manuscript we did not provide a clear schematic of the model and that the twisted substrate made it impossible to clearly distinguish H atoms in the water molecule and at the edge of the substrate. As suggested by the reviewer, we checked the structural model of DFT and provided clearer diagram of the structural model in the revised manuscript.

The following part has been added in the revised manuscript and revised Supplementary information:

Fig. 1 | Theoretical prediction results by DFT calculation. c, HER reaction process for RuNi_{def}/C model under alkaline conditions.

(Please see Fig. 1e in the revised manuscript)

Supplementary Fig. 12 | **a**, Initial state, **b**, transition state and **c**, final state of HER process for RuNi_{def}/C model under alkaline conditions, in which the color yellow indicates H atom in H₂O, and the color white signifies H atom on the edge.

(Please see Fig. 12 in the revised Supplementary information.)

(4) Fig. 1d shows that RuNi_{def}/C possesses a lowest barrier of 0.39 eV, however, it is the endothermic process (~ 0.2 eV), indicating that it is thermodynamically forbidden. Thus, it is difficult to conclude that RuNi_{def}/C is more favorite for water dissociation compared to RuNi_{sub}/C and Ru (002).

Response: We are greatly grateful to the reviewer for your thoughtful consideration. In order to address your valuable comments regarding the lowest barrier of 0.39 eV for RuNi_{def}/C and the thermodynamic significance of this process, we have tried to do our best to explain and cite relevant literature to support it.

According to previous research results (*Nat. commun.* 2021, 12, 6766 and *Angew. Chem. Int. Ed.* 2020, 60, 7234-7244), For the process of water decomposition, the energy barrier of the transition state is theoretically positive, i.e., it is thermodynamically an energy-consuming process that endothermic process. Therefore, the aim of water electrolysis is to minimize the overpotential and to reduce the energy barrier for electrolysis of water, which in turn reduces energy consumption. It is worth noting that in our manuscript, RuNi_{def}/C has the lowest water decomposition transition state energy barrier of 0.39 eV compared to RuNi_{sub}/C and Ru (002), indicating that the RuNi_{def}/C model has an optimal water decomposition potential.

Following up your valuable comments, *the following part and has been added in the revised manuscript to focus on the scientific merits and avoid dispute:*

“It is noteworthy that the transition state energy barriers are positive for all models, which is consistent with the findings reported recently.”

(Please see Line 16-17 of page 7 in the revised manuscript).

(5) The authors noted that “Ru nanocrystals have a high resistance against deactivation because of competitive adsorption of OH intermediates (OH_{ad}) on the Ru and single atoms”, while on $\text{RuNi}_{\text{def}}/\text{C}$ surface only OH adsorbed on Ni single atom has been shown in the current version, for comparison, the other OH adsorption structures or OH and H co-adsorption structures, such as on C or Ru or Ni-Ru bridge, should be shown in supporting information, as well as the adsorption energy.

Response: Thank you for your professional and careful review to improve the quality of our manuscripts. We are sorry that in the previous manuscript we were not very comprehensive and did not provide complete models of OH adsorption structures and OH adsorption energy at various sites. In the revised manuscript, we have considered and provided OH and H adsorption structures and adsorption results for Ni sites, Ru sites, Ni-Ru bridge and Ni_{SAs} anchored on different carbon defect types.

The following part has been added in the revised Supplementary information:

Supplementary Fig. 3 | Theoretical calculation models of RuNi_{def}/C with two typical structures for the graphene edge. a, armchair models, b, zigzag models.

Supplementary Note 1 | The formation energies of the RuNi_{def}/C with armchair-like and zigzag-like graphene edge are both -3.36 eV, indicating that the two models are equally stable.

Supplementary Fig. 4 | Adsorption of OH and H intermediates for RuNi_{def}/C with **a, b** armchair-like and **c, d** zigzag-like graphene edge.

Supplementary Note 2 | The OH* adsorption energies (E_{OH^*}) of the RuNi_{def}/C with armchair-like graphene edge (A-RuNi_{def}/C) and zigzag-like graphene edge (Z-RuNi_{def}/C) are 0.42 eV and 0.39 eV, respectively, indicating that both have similar adsorption capacities for OH*, whereas the ΔG_{H^*} of the corresponding two models are -0.18 and -0.15, indicating that zigzag-like graphene edge has an advantage from this point of view.

Supplementary Fig. 6 | Adsorption of OH intermediates on the Ni site for **a**, Z-Ni_{def}/C and **b**, Z-RuNi_{def}/C.

Supplementary Note 4 | The OH* adsorption energies (E_{OH^*}) on the Ni site for Z-Ni_{def}/C and Z-RuNi_{def}/C are 0.32 eV and 0.39 eV, respectively, indicating that the introduction of Ru site weakened the adsorption of OH at the Ni site due to the interaction between Ru and Ni.

Supplementary Fig. 7 | OH intermediates are adsorbed on the Ru site with the adsorbed Ru located at different distances from Ni: **a**, Ru(002), **b**, near position (RuNi_{def}/C-N), **c**, middle position (RuNi_{def}/C-M) and **d**, far position (RuNi_{def}/C-F).

Supplementary Note 5 | The OH* adsorption energies (E_{OH^*}) on the Ru site for Ru(002), RuNi_{def}/C-N, RuNi_{def}/C-M and RuNi_{def}/C-F are -0.43 eV, -0.15 eV, -0.18 eV and -0.23 eV, respectively, indicating that the adsorption of OH at the Ru site is weakened due to the interaction between Ru and Ni, and the closer the distance between Ru and Ni, the stronger this weakening effect is. It is worth noting that the best model in this manuscript is RuNi_{def}/C-N.

Supplementary Fig. 8 | **a**, Initial and **b**, final states for the OH adsorbed on the Ni-Ru bridge site.

Supplementary Note 6 | When OH adsorbs is adsorbed at the Ni-Ru bridging site, the end state of adsorption remains at the Ru site due to the strong adsorption capacity of Ru. Therefore, the bridging site for OH adsorption is equivalent to the Ru site.

REVIEWER COMMENTS

Reviewer #1 (Remarks to the Author):

The authors have addressed most of my concerns and questions. I appreciate it.

Reviewer #2 (Remarks to the Author):

As the authors answered to the comment 2-2, they failed to present the Ni-Ru bonds. Instead, they toned it down to describe that Ni is atomically dispersed over the catalyst. However, in the revised version, a serious error was found that led to inconsistencies between the material and DFT calculation model. This referee does not believe that the re-revised version of this manuscript can meet the standards of the prestigious Nat comm.

Reviewer #3 (Remarks to the Author):

All my concerns have been answered. It is acceptable.

Responses to Reviewers

We would like to express our gratitude to the reviewers for their valuable comments on our manuscript (NCOMMS-23-43720A). We have carefully considered the latest comments from the reviewers and made revisions to the manuscript. In this latest reply letter, we have listed the comments of the reviewers one by one and numbered them in order. Our response is highlighted in blue, and the revised manuscript and corresponding supplementary information changes are highlighted in yellow.

Following is our careful point-by-point response to all of the reviewers' comments:

Response to Reviewer #1

The authors have addressed most of my concerns and questions. I appreciate it.

Response: Thanks for your positive comments and support for our work. We would like to thank the reviewer again for taking the time to review our manuscript. All of your suggestions are very important and they have been a great guide for me in writing my thesis and in my research, and I hope to learn more from you.

Response to Reviewer #2

As the authors answered to the comment 2-2, they failed to present the Ni-Ru bonds. Instead, they toned it down to describe that Ni is atomically dispersed over the catalyst. However, in the revised version, a serious error was found that led to inconsistencies between the material and DFT calculation model. This referee does not believe that the re-revised version of this manuscript can meet the standards of the prestigious Nat comm.

Response: We are very grateful to the reviewer for his/her professional and critical comments. The reviewers' comments have led to a more rigorous and conscientious approach to our current work and future research, and have been a good guide to my research efforts.

We apologize for the lack of clarity in language presentation and specific data analysis in previous manuscripts, which led to the misinterpretation that the material did not fit the DFT model. The reviewer's comments have made us fundamentally aware of the importance of clarifying whether the Ni-Ru bond exists and the interaction between Ni and Ru in the previous manuscript.

Therefore, we have rigorously addressed and corrected this error in the revised manuscript and Supporting information: **a more rigorous linguistic description and experimental analysis shows that the Ni-Ru bond does not exist and that the Ni_{ISAs} is anchored to C, which has been maintained in our original and revised manuscripts.** Specifically, for the anchored form of Ni single atoms in the UP-RuNi_{ISAs}/C, we attempted to fit it in terms of Ni-O and Ni-C paths and found that the fitting results and parameters were better than those in the previous manuscript, thus the conclusions are more rigorous and reasonable. Combined with the above conclusions, we have explained the interaction of Ni_{ISAs} with C and Ru from different perspectives and by different means of characterization in the newly revised manuscript, in addition to a specific explanation of the corresponding DFT model for the material. The following are specific explanations:

(I) Firstly, we demonstrated the existence of the Ni single atoms in UP-RuNi_{SAs}/C by spherical aberration correction scanning TEM (AC-STEM), and we preliminarily speculated that the single atoms are attributed to Ni single atoms (Ni_{SAs}) by the characterization of AC-STEM, X-ray photoelectron spectroscopy (XPS), and X-ray diffraction (XRD).

(II) Secondly, we mainly apply X-ray absorption spectroscopy (XAS) to demonstrate the anchoring form of Ni_{SAs} and the interaction of elements in the UP-RuNi_{SAs}/C catalyst. **In this part we explain in detail that the Ni-Ru bond does not exist and the Ni Ni_{SAs} is anchored only to C.**

(III) Next, we must apologize for our negligence in misleading you into thinking that the material did not match the calculated model by not providing a clear DFT structural model that presented a comprehensive and correct view of the material. Below and in the revised manuscript, we have corrected this error by presenting a clearer view (top view) of the DFT models to reflect the fact that Ni-C is bonded and Ru-Ni is not bonded in the RuNi_{def}/C model. In addition, in the revised manuscript, we show all the easily misinterpreted unclear computational models in a new light, in order to observe more clearly the connections between the elements in the models.

(IV) Finally, to address your concerns about the effect of Ru-Ni bonding on material properties to verify the superiority of the RuNi_{def}/C model (Ru and Ni are not bonded). We compared the reaction energy barriers of the RuNi alloy model (Ru and Ni bond) with those of RuNi_{def}/C, and ultimately found that the water decomposition energy barrier of RuNi_{def}/C is much lower than that of RuNi alloy. **Particularly, the distance between the Ni and the Ru of RuNi_{def}/C is about 3 Å, whereas the distance between the Ni and Ru atoms in RuNi alloy (where Ni-Ru bonds are present) is about 2 Å, which also confirms that the Ni-Ru bonds are absent in RuNi_{def}/C DFT model.**

The following part has been added in the revised manuscript and revised Supplementary information:

(I)

Supplementary Fig. 21 | Slow scanning XRD patterns of UP-RuNiSAs/C.

Supplementary Note 16 | Slow scanning XRD patterns of the UP-RuNiSAs/C catalyst only detected the partial diffraction peak of the Ru nanocrystal.

(Please see Fig. 21 in the revised Supplementary information.)

“Particularly, the slow scanning XRD pattern of UP-RuNiSAs/C shows diffraction peaks of Ru nanocrystals (PDF no.06-0633) and there is no diffraction signal of Ni, which was initially thought to be a weak content of Ni or the presence of NiSAs.”

(Please see Line 14-15 of Page 9 and Line 1 of Page 10 in the revised manuscript.)

Supplementary Fig. 23 | ICP-OES analysis of UP-RuNiSAs/C.

(Please see Fig. 23 in the revised Supplementary information.)

“ICP-OES accurately analyzed the Ru and Ni contents in the catalysts, revealing atomic ratios of 92.32/7.68 for Ru/Ni in UP-RuNiSAs/C”

(Please see Line 19-21 of Page 10 in the revised manuscript.)

Fig. 3 | Physical characterization and morphology of UP-RuNiSAs/C. **e** and **f**, Spherical aberration correction STEM, in which the bright spots highlighted by pink circles are ascribed to Ni single atoms. **g**, STEM-EDS mapping of UP-RuNiSAs/C.

(Please see Fig. 3e-g in the revised manuscript.)

“NiSAs and Ru nanocrystals can be accurately distinguished and identified using spherical aberration correction scanning TEM (AC-STEM). Fig. 3e revealed the

uniform dispersion of bright spots, indicating the uniform distribution of high-contrast single atoms on carbon fiber paper (CFP). Additionally, higher-resolution AC-TEM enable the observation of the scattered bright spots, while higher brightness Ru crystals can be seen due to the larger atomic number and the Ru (002) surface can be exposed (Fig. 3f). We can preliminarily speculate that the scattered bright spots around the Ru crystal are Ni_{SAs}, which was further validated through STEM-EDS. From Fig. 3g, it can be observed that Ru element is uniformly dispersed at the position of the nanocrystal, while Ni element is uniformly dispersed around the substrate and Ru.”

(Please see Line 9-18 of Page 10 in the revised manuscript.)

Fig. 4 | Characterizations on chemical states and fine structures. f, High-resolution XPS spectra of Ni 2p for UP-Ni/C and UP-RuNi_{SAs}/C catalysts.

(Please see Fig. 4f in the revised manuscript.)

“The fitted Ni 2p spectrum of the UP-RuNi_{SAs}/C (Fig. 4f) shows the valences of Ni²⁺ at 856.1 eV and 873.9 eV due to surface oxidation. Apparently, the Ni⁰ peak was not exhibited in UP-RuNi_{SAs}/C, which is consistent with the AC-STEM results that Ni is present in the form of single atoms.”

(Please see Line 10-13 of Page 14 in the revised manuscript.)

(II)

Fig. 4 | Characterizations on chemical states and fine structures. a, Ni K-edge XANES of UP-RuNiSAs/C, NiO, Ni₂O₃ and Ni foil. **b,** Fourier transforms EXAFS spectra at the Ni K-edge of UP-RuNiSAs/C, NiO, Ni₂O₃ and Ni foil.

(Please see Fig. 4a-b in the revised manuscript.)

“Compared with the NiO, Ni₂O₃ and Ni-foil, the Ni K-edge X-ray absorption near-edge structure (XANES) spectra for UP-RuNiSAs/C was markedly different with the edge of the Ni foil and NiO, with the white line between them, confirming its distinctive coordination structure (Fig. 4a).”

(Please see Line 12-15 of Page 11 in the revised manuscript.)

“As exhibited in Fig. 4b for the Fourier transformed (FT) k²-weighted extended X-ray absorption fine structure (EXAFS) spectrum of UP-RuNiSAs/C, **no metal bonding peaks were found at 2-3 Å compared with NiO, Ni₂O₃ and Ni-foil, corroborating that the Ni is atomically dispersed and the absence of metal-metal coordination of NiSAs in UP-RuNiSAs/C.**”

(Please see Line 19-20 of Page 11 and Line 10-12 of Page 12 in the revised manuscript.)

Fig. 4 | Characterizations on chemical states and fine structures. c, Wavelet transforms for the k^2 -weighted EXAFS signals of UP-RuNiSAs/C, NiO, and Ni foil.

(Please see Fig. 4c in the revised manuscript.)

“For UP-RuNiSAs/C, no Ni-Ni WT signal was observed in Fig. 4c unlike the Ni foil and NiO at $\sim 8.2 \text{ \AA}^{-1}$. In contrast, a Ni coordination with a non-metallic element was observed at $\sim 6.0 \text{ \AA}^{-1}$. Considering that the substrate of the catalyst is carbon, we speculate that it may be a Ni-C coordination.”

(Please see Line 2-5 of Page 13 in the revised manuscript.)

Supplementary Fig. 29 | a, b, c, R space and d, e, f, inverse FT-EXAFS fitting results of Ni K-edge for a, d, Ni foil, b, e, NiO and c, f, UP-RuNiSAs/C.

(Please see Fig. 29 in the revised Supplementary information.)

Supplementary Table 5 | EXAFS fitting parameters at the Ni K-edge for various samples ($S_0^2=1.0$).

Sample	Shell	N ^a	R(Å) ^b	$\sigma^2(\text{Å}^2)^c$	S_0^2	$\Delta E_0(\text{eV})^d$	R factor
Ni foil	Ni-Ni	12	2.48	0.0078	1.0	6.30	0.0103
NiO	Ni-O	6	2.09	0.0076	1.0	2.84	0.0059
	Ni-Ni	12	2.96	0.0064	1.0	9.97	
UP-RuNiSAs/C	Ni-C	3	2.00	0.0061	1.0	6.42	0.0125
	Ni-O	0.4	2.14	0.0075	1.0	5.45	

Supplementary Note 27 | ^aCN, coordination number; ^bR, distance between absorber and backscatter atoms; ^c σ^2 , Debye-Waller factor to account for both thermal and structural disorders; ^d ΔE_0 , inner potential correction; R factor indicates the goodness of the fit. According to the experimental EXAFS fit of Ni foil by fixing CN as the known crystallographic value. Fitting range: $3.0 \leq k (\text{Å}^{-1}) \leq 11.8$ and $1.0 \leq R (\text{Å}) \leq 3.0$ (Ni foil); $3.0 \leq k (\text{Å}^{-1}) \leq 11.8$ and $1.0 \leq R (\text{Å}) \leq 3.0$ (NiO); $3.0 \leq k (\text{Å}^{-1}) \leq 11.2$ and $1.0 \leq R (\text{Å}) \leq 3.1$ (UP-RuNiSAs/C).

(Please see Table S5 in the revised Supplementary information.)

Supplementary Table 5 | EXAFS fitting parameters at the Ni K-edge for various samples ($S_0^2=1.0$).

Sample	Shell	N ^a	R(Å) ^b	$\sigma^2(\text{Å}^2)$ ^c	S_0^2	$\Delta E_0(\text{eV})$ ^d	R factor
Ni foil	Ni-Ni	12	2.48	0.0078	1.0	6.30	0.0103
NiO	Ni-O	6	2.09	0.0076	1.0	2.84	0.0059
	Ni-Ni	12	2.96	0.0064	1.0	9.97	
UP-RuNi _{SAs} /C	Ni-C	3	2.01	0.0064	1.0	10.7	0.0149
	Ni-O	0.4	2.08	0.0093	1.0	10.7	
	Ni/Ru	1	2.78	0.0067	1.0	8.6	

Supplementary Note 27 | ^aCN, coordination number; ^bR, distance between absorber and backscatter atoms; ^c σ^2 , Debye-Waller factor to account for both thermal and structural disorders; ^d ΔE_0 , inner potential correction; R factor indicates the goodness of the fit. According to the experimental EXAFS fit of Ni foil by fixing CN as the known crystallographic value. Fitting range: $3.0 \leq k (\text{Å}^{-1}) \leq 11.8$ and $1.0 \leq R (\text{Å}) \leq 3.0$ (Ni foil); $3.0 \leq k (\text{Å}^{-1}) \leq 11.8$ and $1.0 \leq R (\text{Å}) \leq 3.0$ (NiO).; $3.0 \leq k (\text{Å}^{-1}) \leq 11.2$ and $1.0 \leq R (\text{Å}) \leq 3.1$ (UP-RuNi_{SAs}/C).

(Please see Table S5 in the previous Supplementary information.)

Special note: In order to prove more rigorously that there is no Ni-Ru coordination in the material, We re-performed EXAFS fitting on the Ni-K edge, and the results show that **the EXAFS fitting parameters of the Ni-C/O coordination outperform the previous Ni-C/O and Ni-Ru coordination (as shown in the Table S5 of previous and revised Supplementary information)**, which verifies the speculation of Fig. 4b that the absence of metal-metal coordination in UP-RuNi_{SAs}/C and the Ni_{SAs} anchored to C.

“We verified the above supposition using the EXAFS fitting parameters at the Ni K-edge listed in Supplementary Fig. 29 and Supplementary Table 5. As expected, the coordination structure of Ni-C obtained by fitting the Ni_{ISAS} path is the optimal one and the coordination number (CN) is larger than that of Ni-O (which caused by surface oxidation). **It is worth noting that, as inferred from the absence of Ni-Ru bond in Fig. 4b and the EXAFS fitting in Supplementary Table 5, there is no bonding between Ni and Ru.** The above results confirm that Ni is anchored to carbon in UP-RuNi_{ISAS}/C.”

(Please see Line 5-11 of Page 13 in the revised manuscript.)

Fig. 4 | Characterizations on chemical states and fine structures. d, Ni K-edge XANES of UP-RuNi_{ISAS}/C, Ru foil and RuO₂. **e,** Fourier transform EXAFS spectra of UP-RuNi_{ISAS}/C, Ru foil and RuO₂ at Ru K-edge.

(Please see Fig. 4d-e in the revised manuscript.)

“The Ru K-edge XANES spectra for UP-RuNi_{ISAS}/C showed a similar position and trend to the Ru foil, but with a slightly positive shift in the near edge and white line features (Fig. 4d). This suggests that Ru in the UP-RuNi_{ISAS}/C was slightly oxidized in air. The existence of Ru-Ru coordination for RuNi_{ISAS}/C is evident from the comparison with Ru foil, as UP-RuNi_{ISAS}/C showed distinct Ru-Ru path at around 2.65 Å by the EXAFS results in Fig. 4e.”

(Please see Line 14-19 of Page 13 in the revised manuscript.)

Supplementary Fig. 30 | Wavelet transforms for the k^2 -weighted EXAFS signals of UP-RuNiSAs/C RuO₂ and Ru foil.

(Please see Fig. 30 in the revised Supplementary information.)

“The Ru-Ru WT signal of UP-RuNiSAs/C (Supplementary Fig. 30) also demonstrated Ru-Ru coordination in Ru nanocrystals compared to Ru foil, with Ru-Ru coordination

consistent with Ru foil specimens observed at $\sim 9.8 \text{ \AA}^{-1}$.”

(Please see Line 19-21 of Page 13 in the revised manuscript.)

Supplementary Fig. 31 | **a, b, c**, R space and **d, e, f**, inverse FT-EXAFS fitting results of Ru K-edge for **a, d**, UP-RuNi₃As₄/C, **b, e**, Ru foil and **c, f**, RuO₂.

(Please see Fig. 31 in the revised Supplementary information.)

Supplementary Table 6 | EXAFS fitting parameters at the Ru K-edge for various samples

($S_0^2=1$).

Sample	Path	N^a	$R(\text{\AA})^b$	$\sigma^2(\text{\AA}^2)^c$	$\Delta E_0(\text{eV})^d$	R factor
Ru foil	Ru-Ru	6.00	2.67	0.0034	-4.24	0.0070
RuO ₂	Ru-O	3.57	1.97	0.0027	-0.159	0.0115
	Ru-Ru1	0.94	3.56	0.0055		
	Ru-Ru2	2.72	3.14	0.0040		
UP-RuNi ₃ As ₄ /C	Ru-Ru1	1.81	2.52	0.0029	4.99	0.0100
	Ru-Ru2	1.79	2.71	0.0020		

Supplementary Note 28 | ^aCN, coordination number; ^b R , distance between absorber and backscatter atoms; ^c σ^2 , Debye-Waller factor to account for both thermal and structural disorders; ^d ΔE_0 , inner potential correction; R factor indicates the goodness of the fit. According to the experimental EXAFS fit of Ru foil by fixing CN as the known crystallographic value. Fitting

range: $2.5 \leq k (\text{\AA}^{-1}) \leq 11.6$ and $1.0 \leq R (\text{\AA}) \leq 2.3$ (Ru foil).; $3.0 \leq k (\text{\AA}^{-1}) \leq 11.3$ and $1.0 \leq R (\text{\AA}) \leq 3.7$ (RuO₂). $3.0 \leq k (\text{\AA}^{-1}) \leq 12.7$ and $1.3 \leq R (\text{\AA}) \leq 3.3$ (UP-RuNi₅As/C).

(Please see Table S6 in the revised Supplementary information.)

Sample	Path	Na	R(Å) ^b	$\sigma^2(\text{\AA}^2)$ ^c	$\Delta E_0(\text{eV})$ ^d	R factor
Ru foil	Ru-Ru	6.00	2.67	0.0034	-4.24	0.0070
RuO ₂	Ru-O	3.57	1.97	0.0027	-0.159	0.0115
	Ru-Ru1	0.94	3.56	0.0055		
	Ru-Ru2	2.72	3.14	0.0040		
UP-RuNi ₅ As/C	Ru-Ru	4.04	2.66	0.0033	-5.47	0.0113
	Ru-Ni	3.72	2.58	0.0224		

Note: ^aCN, coordination number; ^bR, distance between absorber and backscatter atoms; ^c σ^2 , Debye-Waller factor to account for both thermal and structural disorders; ^d ΔE_0 , inner potential correction; R factor indicates the goodness of the fit. According to the experimental EXAFS fit of Ru foil by fixing CN as the known crystallographic value. Fitting range: $2.5 \leq k (\text{\AA}^{-1}) \leq 11.6$ and $1.0 \leq R (\text{\AA}) \leq 2.3$ (Ru foil).; $3.0 \leq k (\text{\AA}^{-1}) \leq 11.3$ and $1.0 \leq R (\text{\AA}) \leq 3.7$ (RuO₂). $3.0 \leq k (\text{\AA}^{-1}) \leq 12.7$ and $1.3 \leq R (\text{\AA}) \leq 3.3$ (UP-RuNi₅As/C).

(Fitting results considering Ru-Ni coordination in UP-RuNi₅As/C)

Special note: In addition, in order to further prove that the Ru-Ru path of Ru-K edge EXAFS fitting is optimal, we try to introduce Ru-Ni coordination, and the results show that **EXAFS fitting parameters of the Ru-Ni coordination is not superior to the Ru-Ru coordination. Therefore, in combination with the coordination of Ni elements, we further confirmed the presence of Ru in the form of Ru-Ru coordination in UP-RuNi₅As/C.**

“The Ru-Ru coordination structure of the Ru element in RuNi₅As/C is further evidenced by the EXAFS fitting parameters at the Ru K-edge in Supplementary Figure 31 and Supplementary Table 6, and the fitting parameters revealed the decrease in the average coordination number of Ru in UP-RuNi₅As/C compared to Ru foil, possibly due to the smaller size of Ru nanoparticles in the UP-RuNi₅As/C catalysts (nano effect).”

(Please see Line 21-22 of Page 13 and Line 1-4 of Page 14 in the revised manuscript.)

“It is preliminarily concluded that Ni single atoms modified Ru nanoparticles via carbon bridging (UP-RuNi_{SAs}/C) have been successfully synthesized, in which Ni single atoms are anchored to the substrate carbon.”

(Please see Line 5-7 of Page 14 in the revised manuscript.)

(III)

Supplementary Fig. 1 | Theoretical calculation models. a, Ru (002), b, Ni (111), c, Ni_{sub}/C and d, Ni_{def}/C.

(Please see Fig. 1 in the revised Supplementary information.)

“To model the structure of Ru nanocrystals, we focused on the Ru (002) surface of the hexagonal Ru structure (Supplementary Fig. 1a). Subsequently, Ni structures at various scales including Ni (111) index surface of cubic Ni (Supplementary Fig. 1b), Ni single atom anchoring on the substitutional (bulk) site of graphite carbon (Ni_{sub}/C) (Supplementary Fig. 1c) and Ni single atom anchoring on the edge (defect) site of graphite carbon (Ni_{def}/C) (Supplementary Fig. 1d) were constructed to investigate their effects on Ru nanocrystals.”

(Please see Line 15-20 of Page 4 in the revised manuscript.)

Supplementary Fig. 3 | Theoretical calculation models of RuNi_{def}C with two typical structures for the graphene edge. a, armchair models, b, zigzag models.

Supplementary Note 1 | The formation energies of the RuNi_{def}C with armchair-like and zigzag-like graphene edge are both -3.36 eV, indicating that the two models are equally stable.

(Please see Fig. 3 in the revised Supplementary information.)

“It is important to note that we performed an initial screening of the substrate carbon model in terms of formation energy, intermediate adsorption, and water decomposition barriers, respectively (Supplementary Fig. 3-8). We ultimately determined that the most stable and suitable Ni_{SAs}-anchored carbon defect is the zigzag-like graphene edge (Supplementary Fig. 3b).”

(Please see Line 20-22 of Page 4 and Line 1-2 of Page 5 in the revised manuscript.)

Supplementary Table 1 | Adsorption energy of various intermediate adsorption on active sites for HER corresponding to different models.

Model	E _{OH*} (eV)	E _{H*} (eV)
Ru (002)	-0.43	0.53
Ni (111)	-0.25	0.71

Ni _{sub}	0.29	0.72
Ni _{def}	0.32	0.55
RuNi _{sub} /C	0.35	0.24
RuNi _{def} /C	-0.15	0.15

(Please see Table S1 in the revised Supplementary information.)

“The adsorption of OH* was precisely investigated firstly through the calculating the adsorption energy (E_{OH^*}) for different models in Fig. 1a and Supplementary Table 1. Notably, Ni_{def}/C has the weakest adsorption of OH* ($E_{OH^*}=0.32$ eV) compared to the Ni (111) of -0.25 eV and Ni_{sub}/C of 0.29 eV, and Ni_{def}/C has a superior free energy of H* (E_{H^*}) compared to Ni_{sub}/C, suggesting that Ni_{SAs} is more favorable for weakening OH* adsorption on Ru sites relative to metal Ni, resulting in the optimal OH* adsorption energy for Ru (002) in RuNi_{def}/C ($E_{OH^*}=-0.15$ eV) for exposure of active sites. **In particular, in the RuNi_{def}/C model for this work (Supplementary Fig. 3b), Ni_{SAs} occupying the defective carbon sites, whereas there is no bonding between Ru and Ni.**”

(Please see Line 2-10 of Page 5 in the revised manuscript.)

The following part has been added in the revised manuscript and revised Supplementary information for different model structures with new perspectives:

Fig. 1 | Theoretical prediction results by DFT calculation. e, HER reaction process for RuNi_{def}/C model under alkaline conditions.

(Please see Fig. 1e in the revised manuscript.)

Supplementary Fig. 12 | a, Initial state, b, transition state and c, final state of HER reaction process for RuNi_{def}/C model under alkaline conditions, in which the color yellow indicates H atom in H₂O, and the color white signifies H atom on the edge.

Supplementary Note 8 | As can be seen from the Supplementary Fig. 4, the water molecules are first adsorbed on Ru sites. In the transition state, OH intermediates are preferentially adsorbed on Ni sites.

(Please see Fig. 12 in the revised Supplementary information.)

Supplementary Fig. 2 | Theoretical calculation model of RuNi_{sub}/C.

(Please see Fig. 2 in the revised Supplementary information.)

Supplementary Fig. 4 | Adsorption of OH and H intermediates for RuNi_{def}/C with **a, b** armchair-like and **c, d** zigzag-like graphene edge.

Supplementary Note 2 | The OH* adsorption energies (E_{OH^*}) of the RuNi_{def}/C with armchair-like graphene edge (A-RuNi_{def}/C) and zigzag-like graphene edge (Z-RuNi_{def}/C) are 0.42 eV and 0.39 eV, respectively, indicating that both have similar adsorption capacities for OH*, whereas the ΔG_{H^*} of the corresponding two models are -0.18 and -0.15, indicating that zigzag-like graphene edge has an advantage from this point of view.

(Please see Fig. 4 in the revised Supplementary information.)

Supplementary Fig. 6 | Adsorption of OH intermediates on the Ni site for **a**, Z-Ni_{def}/C and **b**, Z-RuNi_{def}/C.

Supplementary Note 4 | The OH* adsorption energies (E_{OH^*}) on the Ni site for Z-Ni_{def}/C and Z-RuNi_{def}/C are 0.32 eV and 0.42 eV, respectively, indicating that the introduction of Ru site weakened the adsorption of OH at the Ni site due to the interaction between Ru and Ni.

(Please see Fig. 6 in the revised Supplementary information.)

Supplementary Fig. 7 | OH intermediates is adsorbed on the Ru site with the adsorbed Ru is located at different distances from Ni : **a**, Ru (002), **b**, near position (RuNi_{def}/C-N), **c**, middle position (RuNi_{def}/C-M) and **d**, far position (RuNi_{def}/C-F).

Supplementary Note 5 | The OH* adsorption energies (E_{OH^*}) on the Ru site for Ru (002), RuNi_{def}/C-N, RuNi_{def}/C-M and RuNi_{def}/C-F are -0.43 eV, -0.15 eV, -0.18 eV and -0.23 eV, respectively, indicating that the adsorption of OH at the Ru site is weakened due to the interaction between Ru and Ni, and the closer the distance between Ru and Ni, the stronger this weakening effect is. It is worth noting that the best model in this manuscript is RuNi_{def}/C-N (RuNi_{def}/C).

(Please see Fig. 7 in the revised Supplementary information.)

Supplementary Fig. 8 | **a**, Initial and **b**, final states for the OH adsorbed on the Ni-Ru bridge site.

Supplementary Note 6 | When OH adsorbs is adsorbed at the Ni-Ru bridging site, the end state of adsorption remains at the Ru site due to the strong adsorption capacity of Ru. Therefore, the bridging site for OH adsorption is equivalent to the Ru site.

(Please see Fig. 8 in the revised Supplementary information.)

Supplementary Fig. 11 | **a**, Initial state, **b**, transition state and **c**, final state of HER reaction process for Ru (002) model under alkaline conditions.

(Please see Fig. 11 in the revised Supplementary information.)

(IV)

Supplementary Fig. 13 | a, Initial state, b, transition state and c, final state of HER process for RuNi alloy model under alkaline conditions.

Supplementary Note 9 | In order to compare with the RuNi_{def}/C model, the RuNi alloy with Ni partially replacing Ru crystals (002) was modeled and its water decomposition energy barrier was calculated. **Particularly, the distance between the Ni and the Ru of RuNi_{def}/C is about 3 Å, whereas the distance between the Ni and Ru atoms in RuNi alloy (where Ni-Ru bonds are present) is about 2 Å, which also confirms that the Ni-Ru bonds are absent in RuNi_{def}/C DFT model.**

(Please see Fig. 13 in the revised Supplementary information.)

Supplementary Fig. 14 | Reaction coordinates for HER process for RuNi alloy.

Supplementary Note 10 | The RuNi alloy has a high transition state energy barrier 0.7 eV, even higher than 0.52 eV of pure Ru (002).

(Please see Fig. 14 in the revised Supplementary information.)

Fig. 1 | Theoretical prediction results by DFT calculation. f, Reaction coordinates for HER process for Ru (002), RuNi_{sub}/C and RuNi_{def}/C models.

(Please see Fig. 1f in the revised manuscript.)

“In addition, the RuNi alloy model has a higher transition state energy barrier of 0.7 eV (Supplementary Fig. 13-14), which does not reflect an advantage over pure Ru (0.52 eV) and RuNi_{def}/C (0.39 eV), Further proving the unique modification effect of Ni single atom anchored defect carbon on Ru nanocrystals”

(Please see Line 16-19 of Page 7 in the revised manuscript.)

Response to Reviewer #3

All my concerns have been answered. It is acceptable.

Response: Thank you again for your professional comments and valuable suggestions about our work. Your suggestions have helped us to improve the quality of our manuscript and will guide us in our future work.

REVIEWERS' COMMENTS

Reviewer #2 (Remarks to the Author):

The manuscript has been revised by fully reflecting the reviewers' concerns. The current version looks publishable as is.